# SKILL-BASED META-REINFORCEMENT LEARNING

**Taewook Nam[1], Shao-Hua Sun[2], Karl Pertsch[2], Sung Ju Hwang[1,3], Joseph J. Lim[1†,4‡]**
Korea Advanced Institute of Science and Technology[1]
University of Southern California[2], AITRICS[3], Naver AI Lab[4]
namsan@kaist.ac.kr,{shaohuas,pertsch}@usc.edu,
sjhwang82@kaist.ac.kr,joe.lim@kaist.ac.kr

## ABSTRACT

While deep reinforcement learning methods have shown impressive results in robot learning, their sample inefficiency makes the learning of complex, long-horizon behaviors with real robot systems infeasible. To mitigate this issue, meta-reinforcement learning methods aim to enable fast learning on novel tasks by learning how to learn. Yet, the application has been limited to short-horizon tasks with dense rewards. To enable learning long-horizon behaviors, recent works have explored leveraging prior experience in the form of offline datasets without reward or task annotations. While these approaches yield improved sample efficiency, millions of interactions with environments are still required to solve complex tasks. In this work, we devise a method that enables meta-learning on long-horizon, sparse-reward tasks, allowing us to solve unseen target tasks with orders of magnitude fewer environment interactions. Our core idea is to leverage prior experience extracted from offline datasets during meta-learning. Specifically, we propose to (1) extract reusable skills and a skill prior from offline datasets, (2) meta-train a high-level policy that learns to efficiently compose learned skills into long-horizon behaviors, and (3) rapidly adapt the meta-trained policy to solve an unseen target task. Experimental results on continuous control tasks in navigation and manipulation demonstrate that the proposed method can efficiently solve long-horizon novel target tasks by combining the strengths of meta-learning and the usage of offline datasets, while prior approaches in RL, meta-RL, and multi-task RL require substantially more environment interactions to solve the tasks.

## 1 INTRODUCTION

In recent years, deep reinforcement learning methods have achieved impressive results in robot learning (Gu et al., 2017; Andrychowicz et al., 2020; Kalashnikov et al., 2021). Yet, existing approaches are sample inefficient, thus rendering the learning of complex behaviors through trial and error learning infeasible, especially on real robot systems. In contrast, humans are capable of effectively learning a variety of complex skills in only a few trials. This can be greatly attributed to our ability to learn how to learn new tasks quickly by efficiently utilizing previously acquired skills.

Can machines likewise learn to how to learn by efficiently utilizing learned skills like humans? Meta-reinforcement learning (meta-RL) holds the promise of allowing RL agents to acquire novel tasks with improved efficiency by learning to learn from a distribution of tasks (Finn et al., 2017; Rakelly et al., 2019). In spite of recent advances in the field, most existing meta-RL algorithms are restricted to short-horizon, dense-reward tasks. To facilitate efficient learning on long-horizon, sparse-reward tasks, recent works aim to leverage experience from prior tasks in the form of offline datasets without additional reward and task annotations (Lynch et al., 2020; Pertsch et al., 2020; Chebotar et al., 2021). While these methods can solve complex tasks with substantially improved sample efficiency over methods learning from scratch, millions of interactions with environments are still required to acquire long-horizon skills.

---

[†]Work done while at USC

[‡]AI advisor at Naver AI Lab
Project page: https://namsan96.github.io/SiMPL

In this work, we aim to take a step towards combining the capabilities of *both* learning how to quickly learn new tasks while *also* leveraging prior experience in the form of unannotated offline data (see Figure 1). Specifically, we aim to devise a method that enables meta-learning on complex, long-horizon tasks and can solve unseen target tasks with orders of magnitude fewer environment interactions than prior works.

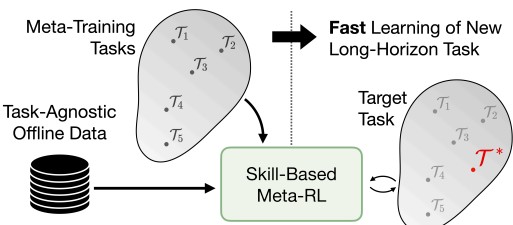

We propose to leverage the offline experience by extracting reusable *skills* – short-term behaviors that can be composed to solve unseen long-horizon tasks. We employ a hierarchical meta-learning scheme in which we meta-train a high-level policy to learn how to quickly reuse

Figure 1: We propose a method that jointly leverages (1) a large offline dataset of prior experience collected across many tasks without reward or task annotations and (2) a set of meta-training tasks to learn how to quickly solve unseen long-horizon tasks. Our method extracts reusable skills from the offline dataset and meta-learn a policy to quickly use them for solving new tasks.

the extracted skills. To efficiently explore the learned skill space during meta-training, the high-level policy is guided by a skill prior which is also acquired from the offline experience data.

We evaluate our method and prior approaches in deep RL, skill-based RL, meta-RL, and multi-task RL on two challenging continuous control environments: maze navigation and kitchen manipulation, which require long-horizon control and only provides sparse rewards. Experimental results show that our method can efficiently solve unseen tasks by exploiting meta-learning tasks and offline datasets, while prior approaches require substantially more samples or fail to solve the tasks.

In summary, the main contributions of this paper are threefold:

- To the best of our knowledge, this is the first work to combine meta-reinforcement learning algorithms with task-agnostic offline datasets that do not contain reward or task annotations.
- We propose a method that combines meta-learning with offline data by extracting learned skills and a skill prior as well as meta-learning a hierarchical skill policy regularized by the skill prior.
- We empirically show that our method is significantly more efficient at learning long-horizon sparse-reward tasks compared to prior methods in deep RL, skill-based RL, meta-RL, and multi-task RL.

## 2 RELATED WORK

**Meta-Reinforcement Learning.** Meta-RL approaches (Duan et al., 2016; Wang et al., 2017; Finn et al., 2017; Yu et al., 2018; Rothfuss et al., 2019; Gupta et al., 2018; Vuorio et al., 2018; Nagabandi et al., 2019; Clavera et al., 2019; 2018; Rakelly et al., 2019; Vuorio et al., 2019; Yang et al., 2019; Zintgraf et al., 2019; Humplik et al., 2019; Zintgraf et al., 2020; Liu et al., 2021) hold the promise of allowing learning agents to quickly adapt to novel tasks by learning to learn from a distribution of tasks. Despite the recent advances in the field, most existing meta-RL algorithms are limited to short-horizon and dense-reward tasks. In contrast, we aim to develop a method that can meta-learn to solve long-horizon tasks with sparse rewards by leveraging offline datasets.

**Offline datasets.** Recently, many works have investigated the usage of offline datasets for agent training. In particular, the field of *offline reinforcement learning* (Levine et al., 2020; Siegel et al., 2020; Kumar et al., 2020; Yu et al., 2021) aims to devise methods that can perform RL fully offline from pre-collected data, without the need for environment interactions. However, these methods require target task reward annotations on the offline data for every new tasks that should be learned. These reward annotations can be challenging to obtain, especially if the offline data is collected from a diverse set of prior tasks. In contrast, our method is able to leverage offline datasets without any reward annotations.

**Offline Meta-RL.** Another recent line of research aims to *meta-learn* from static, pre-collected datasets including reward annotations (Mitchell et al., 2021; Pong et al., 2021; Dorfman et al., 2021). After meta-training with the offline datasets, these works aim to quickly adapt to a new task with only a small amount of data from that new task. In contrast to the aforementioned offline RL methods

these works aim to adapt to *unseen* tasks and assume access to only *limited data* from the new tasks. However, in addition to reward annotations, these approaches often require that the offline training data is split into separate datasets for each training tasks, further limiting the scalability.

**Skill-based Learning.** An alternative approach for leveraging offline data that does not require reward or task annotations is through the extraction of skills – reusable short-horizon behaviors. Methods for skill-based learning recombine these skills for learning unseen target tasks and converge substantially faster than methods that learn from scratch (Lee et al., 2018; Hausman et al., 2018; Sharma et al., 2020; Sun, 2022). When trained from diverse datasets these approaches can extract a wide repertoire of skills and learn complex, long-horizon tasks (Merel et al., 2020; Lynch et al., 2020; Pertsch et al., 2020; Ajay et al., 2021; Chebotar et al., 2021; Pertsch et al., 2021). Yet, although they are more efficient than training from scratch, they still require a large number of environment interactions to learn a new task. Our method instead combines skills extracted from offline data with meta-learning, leading to significantly improved sample efficiency.

## 3 PROBLEM FORMULATION AND PRELIMINARIES

Our approach builds on prior work for meta-learning and learning from offline datasets and aims to combine the best of both worlds. In the following we will formalize our problem setup and briefly summarize relevant prior work.

**Problem Formulation.** Following prior work on learning from large offline datasets (Lynch et al., 2020; Pertsch et al., 2020; 2021), we assume access to a dataset of state-action trajectories $\mathbf{D} = \{s_t, a_t, ...\}$ which is collected either across a wide variety of tasks or as "play data" with no particular task in mind. We thus refer to this dataset as *task-agnostic*. With a large number of data collection tasks, the dataset covers a wide variety of behaviors and can be used to accelerate learning on diverse tasks. Such data can be collected at scale, *e.g.* through autonomous exploration (Hausman et al., 2018; Sharma et al., 2020; Dasari et al., 2019), human teleoperation (Schaal et al., 2005; Gupta et al., 2019; Mandlekar et al., 2018; Lynch et al., 2020), or from previously trained agents (Fu et al., 2020; Gulcehre et al., 2020). We additionally assume access to a set of meta-training tasks $\mathbf{T} = \{\mathcal{T}_1, \ldots, \mathcal{T}_N\}$, where each task is represented as a Markov decision process (MDP) defined by a tuple $\{\mathcal{S}, \mathcal{A}, \mathcal{P}, r, \rho, \gamma\}$ of states, actions, transition probability, reward, initial state distribution, and discount factor.

Our goal is to leverage both, the offline dataset $\mathbf{D}$ and the meta-training tasks $\mathbf{T}$, to accelerate the training of a policy $\pi(a|s)$ on a target task $\mathcal{T}^*$ which is also represented as an MDP. Crucially, we do not assume that $\mathcal{T}^*$ is a part of the set of training tasks $\mathbf{T}$, nor that $\mathbf{D}$ contains demonstrations for solving $\mathcal{T}^*$. Thus, we aim to design an algorithm that can leverage offline data and meta-training tasks for learning how to quickly compose known skills for solving an unseen target task. Next, we will describe existing approaches that *either* leverage offline data *or* meta-training tasks to accelerate target task learning. Then, we describe how our approach takes advantage of the best of both worlds.

**Skill-based RL.** One successful approach for leveraging task-agnostic datasets for accelerating the learning of unseen tasks is though the transfer of reusable *skills*, *i.e.* short-horizon behaviors that can be composed to solve long-horizon tasks. Prior work in skill-based RL called Skill-Prior RL (SPiRL, Pertsch et al. (2020)) proposes an effective way to implement this idea. Specifically, SPiRL uses a task-agnostic dataset to learns two models: (1) a skill policy $\pi(a|s, z)$ that decodes a latent skill representation $z$ into a sequence of executable actions and (2) a prior over latent skill variables $p(z|s)$ which can be leveraged to guide exploration in skill space. SPiRL uses these skills for learning new tasks efficiently by training a high-level skill policy $\pi(z|s)$ that acts over the space of learned skills instead of primitive actions. The target task RL objective extends Soft Actor Critic (SAC, Haarnoja et al. (2018)), a popular off-policy RL algorithm, by guiding the high-level policy with the learned skill prior:

$$\max_\pi \sum_t \mathbb{E}_{(s_t, z_t) \sim \rho_\pi} \big[ r(s_t, z_t) - \alpha D_{\mathrm{KL}}\big(\pi(z|s_t), p(z|s_t)\big)\big]. \tag{1}$$

Here $D_{\mathrm{KL}}$ denotes the Kullback-Leibler divergence between the policy and skill prior, and $\alpha$ is a weighting coefficient.

**Off-Policy Meta-RL.** Rakelly et al. (2019) introduced an off-policy meta-RL algorithm called probabilistic embeddings for actor-critic RL (PEARL) that leverages a set of training tasks $\mathbf{T}$ to

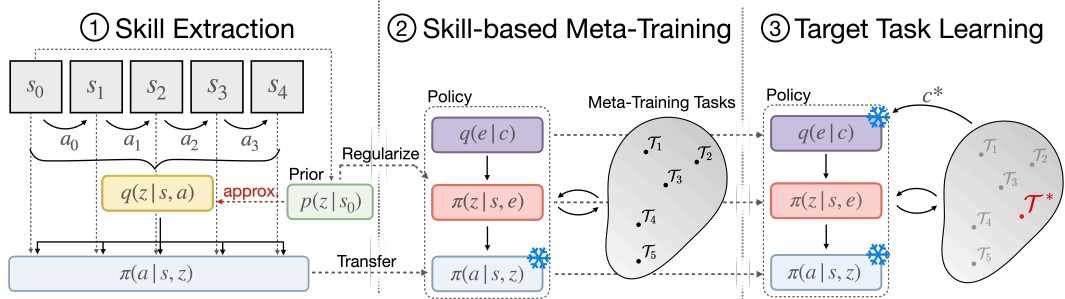

Figure 2: **Method Overview.** Our proposed skill-based meta-RL method has three phases. **(1) Skill Extraction**: learns reusable skills from snippets of task-agnostic offline data through a skill extractor (**yellow**) and low-level skill policy (**blue**). Also trains a prior distribution over skill embeddings (**green**). **(2) Skill-based Meta-training**: Meta-trains a high-level skill policy (**red**) and task encoder (**purple**) while using the pre-trained low-level policy. The pre-trained skill prior is used to regularize the high-level policy during meta-training and guide exploration. **(3) Target Task Learning**: Leverages the meta-trained hierarchical policy for quick learning of an unseen target task. After conditioning the policy by encoding a few transitions $c^*$ from the target task $\mathcal{T}^*$, we continue fine-tuning the high-level skill policy on the target task while regularizing it with the pre-trained skill prior.

enable quick learning of new tasks. Specifically, PEARL leverages the meta-training tasks for learning a task encoder $q(e|c)$. This encoder takes in a small set of state-action-reward transitions $c$ and produces a task embedding $e$. This embedding is used to condition the actor $\pi(a|s, z)$ and critic $Q(s, a, e)$. In PEARL, actor, critic and task encoder are trained by jointly maximizing the obtained reward and the policy's entropy $\mathcal{H}$ (Haarnoja et al., 2018):

$$\max_{\pi} \mathbb{E}_{\mathcal{T} \sim p_{\mathcal{T}}, e \sim q(\cdot|c^{\mathcal{T}})} \left[ \sum_t \mathbb{E}_{(s_t, a_t) \sim \rho_{\pi|e}} \left[ r_{\mathcal{T}}(s_t, a_t) + \alpha \mathcal{H}\big(\pi(a|s_t, e)\big) \right] \right]. \tag{2}$$

Additionally, the task embedding output of the task encoder is regularized towards a constant prior distribution $p(e)$.

## 4 APPROACH

We propose **Ski**ll-based **M**eta-**P**olicy **L**earning (SiMPL), an algorithm for jointly leveraging offline data as well as a set of meta-training tasks to accelerate the learning of unseen target tasks. Our method has three phases: **(1) skill extraction**: we extract reusable skills and a skill prior from the offline data (Section 4.1), **(2) skill-based meta-training**: we utilize the meta-training tasks to learn how to leverage the extracted skills and skill prior to efficiently solve new tasks (Section 4.2), **(3) target task learning**: we fine-tune the meta-trained policy to rapidly adapt to solve an unseen target task (Section 4.3). An illustration of the proposed method is shown in Figure 2.

### 4.1 SKILL EXTRACTION

To acquire a set of reusable skills from the offline dataset $\mathbf{D}$, we leverage the skill extraction approach proposed in Pertsch et al. (2020). Concretely, we jointly train (1) a skill encoder $q(z|s_{0:K}, a_{0:K-1})$ that embeds an $K$-steps trajectory randomly cropped from the sequences in $\mathbf{D}$ into a low-dimensional skill embedding $z$, and (2) a low-level skill policy $\pi(a_t|s_t, z)$ that is trained with behavioral cloning to reproduce the action sequence $a_{0:K-1}$ given the skill embedding. To learn a smooth skill representation, we regularize the output of the skill encoder with a unit Gaussian prior distribution, and weight this regularization by a coefficient $\beta$ (Higgins et al., 2017):

$$\max_{q, \pi} \mathbb{E}_{z \sim q} \Big[ \underbrace{\prod_{t=0}^{K-1} \log \pi(a_t|s_t, z)}_{\text{behavioral cloning}} - \beta \underbrace{D_{\text{KL}}\big(q(z|s_{0:K}, a_{0:K-1}), \mathcal{N}(0, I)\big)}_{\text{embedding regularization}} \Big]. \tag{3}$$

Additionally, we follow Pertsch et al. (2020) and learn a skill prior $p(z|s)$ that captures the distribution of skills likely to be executed in a given state under the training data distribution. The prior is trained to

match the output of the skill encoder: $\min_p D_{\mathrm{KL}}\big(\lfloor q(z|s_{0:K}, a_{0:K-1})\rfloor, p(z|s_0)\big)$. Here $\lfloor \cdot \rfloor$ indicates that gradient flow is stopped into the skill encoder for training the skill prior.

## 4.2 SKILL-BASED META-TRAINING

We aim to learn a policy that can quickly learn to leverage the extracted skills to solve new tasks. We leverage off-policy meta-RL (see Section 3) to learn such a policy using our set of meta-training tasks **T**. Similar to PEARL (Rakelly et al., 2019), we train a task-encoder that takes in a set of sampled transitions and produces a task embedding $e$. Crucially, we leverage our learned skills by training a task-embedding-conditioned policy over *skills* instead of primitive actions: $\pi(z|s, e)$, thus equipping the policy with a set of useful pre-trained behaviors and reducing the meta-training task to learning how to combine these behaviors instead of learning them from scratch. We find that this usage of offline data through learned skills is crucial for enabling meta-training on complex, long-horizon tasks (see Section 5).

Prior work has shown that the efficiency of RL on learned skill spaces can be substantially improved by guiding the policy with a learned skill prior (Pertsch et al., 2020; Ajay et al., 2021). Thus, instead of regularizing with a maximum entropy objective as done in prior work on off-policy meta-RL (Rakelly et al., 2019), we propose to regularize the meta-training policy with our pre-trained skill prior, leading to the following meta-training objective:

$$\max_\pi \mathbb{E}_{\mathcal{T}\sim p_{\mathcal{T}}, e\sim q(\cdot|c^{\mathcal{T}})}\left[\sum_t \mathbb{E}_{(s_t,z_t)\sim\rho_{\pi|e}}\big[r_{\mathcal{T}}(s_t, z_t) - \alpha D_{\mathrm{KL}}\big(\pi(z|s_t, e), p(z|s_t)\big)\big]\right]. \qquad (4)$$

where $\alpha$ determines the strength of the prior regularization. We automatically tune $\alpha$ via dual gradient descent by choosing a target divergence $\delta$ between policy and prior (Pertsch et al., 2020).

To compute the task embedding $e$, we used multiple different sizes of $c$. We found that we can increase training stability by adjusting the strength of the prior regularization to the size of the conditioning set. Intuitively, when the high-level policy is conditioned on only a few transitions, i.e. when the set $c$ is small, it has only little information about the task at hand and should thus be regularized stronger towards the task-agnostic skill prior. Conversely, when $c$ is large, the policy likely has more information about the target task and thus should be allowed to deviate from the skill prior more to solve the task, i.e. have a weaker regularization strength.

To implement this intuition, we employ a simple approach: we define *two* target divergences $\delta_1$ and $\delta_2$ and associated auto-tuned coefficients $\alpha_1$ and $\alpha_2$ with $\delta_1 < \delta_2$. We regularize the policy using the larger coefficient $\alpha_1$ with small conditioning transition set and otherwise we regularize using the smaller coefficient $\alpha_2$. We found this technique simple yet sufficient in our experiments and leave the investigation of more sophisticated regularization approaches for future work.

## 4.3 TARGET TASK LEARNING

When a target task is given, we aim to leverage the meta-trained policy for quickly learning how to solve it. Intuitively, the policy should first explore different skill options to learn about the task at hand and then rapidly narrow its output distribution to those skills that solve the task. We implement this intuition by first collecting a small set of conditioning transitions $c^*$ from the target task by exploring with the meta-trained policy. Since we have no information about the target task at this stage, we explore the environment by conditioning our pre-trained policy with task embeddings sampled from the task prior $p(e)$. Then, we encode this set of transitions into a target task embedding $e^* \sim q(e|c^*)$. By conditioning our meta-trained high-level policy on this encoding, we can rapidly narrow its skill distribution to skills that solve the given target task: $\pi(z|s, e^*)$.

Empirically, we find that this policy is often already able to achieve high success rates on the target task. Note that only very few interactions with the environment for collecting $c^*$ are required for learning a complex, long-horizon and unseen target task with sparse reward. This is substantially more efficient than prior approaches such as SPiRL that require orders of magnitude more target task interactions for achieving comparable performance.

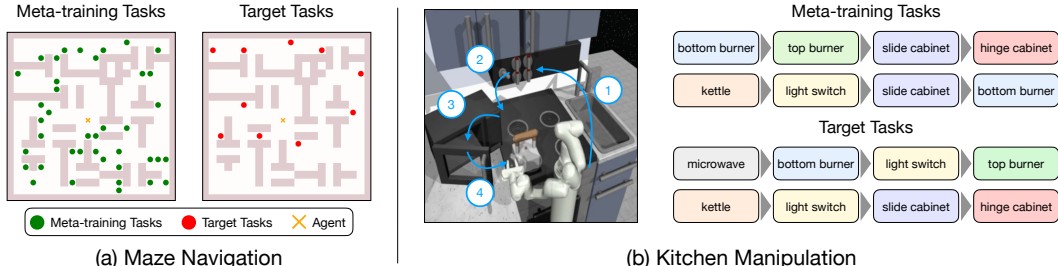

Figure 3: **Environments.** We evaluate our proposed framework in two domains that require the learning of complex, long-horizon behaviors from sparse rewards. These environments are substantially more complex than those typically used to evaluate meta-RL algorithms. (a) **Maze Navigation**: The agent needs to navigate for hundreds of steps to reach unseen target goals and only receives a binary reward upon task success. (b) **Kitchen Manipulation**: The 7DoF agent needs to execute an unseen sequence of four subtasks, spanning hundreds of time steps, and only receives a sparse reward upon completion of each subtask.

To further improve the performance on the target task, we fine-tune the conditioned policy with target task rewards while guiding its exploration with the pre-trained skill prior[1]:

$$\max_\pi \mathbb{E}_{e^* \sim q(\cdot|c^*)} \left[ \sum_t \mathbb{E}_{(s_t, z_t) \sim \rho_{\pi|e^*}} \left[ r_{\mathcal{T}^*}(s_t, z_t) - \alpha D_{\mathrm{KL}} \big( \pi(z|s_t, e^*), p(z|s_t) \big) \right] \right]. \qquad (5)$$

More implementation details on our method can be found in Section E.

## 5 EXPERIMENTS

Our experiments aim to answer the following questions: (1) Can our proposed method learn to efficiently solve long-horizon, sparse reward tasks? (2) Is it crucial to utilize offline datasets to achieve this? (3) How can we best leverage the training tasks for efficient learning of target tasks? (4) How does the training task distribution affect the target task learning?

### 5.1 EXPERIMENTAL SETUP

We evaluate our approach in two challenging continuous control environments: maze navigation and kitchen manipulation environment, as illustrated in Figure 3. While meta-RL algorithms are typically evaluated on tasks that span only a few dozen time steps and provide dense rewards (Finn et al., 2017; Rothfuss et al., 2019; Rakelly et al., 2019; Zintgraf et al., 2020), our tasks require to learn long-horizon behaviors over hundreds of time steps from sparse reward feedback and thus pose a new challenge to meta-learning algorithms.

#### 5.1.1 MAZE NAVIGATION

**Environment.** This 2D maze navigation domain based on the maze navigation problem in Fu et al. (2020) requires long-horizon control with hundreds of steps for a successful episode and only provides sparse reward feedback upon reaching the goal. The observation space of the agent consists of its 2D position and velocity and it acts via planar, continuous velocity commands.

**Offline Dataset & Meta-training / Target Tasks.** Following Fu et al. (2020) we collect a task-agnostic offline dataset by randomly sampling start-goal locations in the maze and using a planner to generate a trajectory that reaches from start to goal. Note that the trajectories are not annotated with any reward or task labels (*i.e.* which start-goal location is used for producing each trajectory). To generate a set of meta-training and target tasks, we fix the agent's initial position in the center of the maze and sample 40 random goal locations for meta-training and another set of 10 goals for target

---

[1]Other regularization distributions are possible during fine-tuning, e.g. the high-level policy conditioned on task prior samples $p(z|s, e \sim p(e))$ or the target task embedding conditioned policy $p(z|s, e^*)$ *before finetuning*. Yet, we found the regularization with the pre-trained task-agnostic skill prior to work best in our experiments.

tasks. All meta-training and target tasks use the same sparse reward formulation. More details can be found in Section G.1.

### 5.1.2 KITCHEN MANIPULATION

**Environment.** The FrankaKitchen environment of Gupta et al. (2019) requires the agent to control a 7-DoF robot arm via continuous joint velocity commands and complete a sequence of manipulation tasks like opening the microwave or turning on the stove. Successful episodes span 300-500 steps and the agent is only provided a sparse reward signal upon successful completion of a subtask.

**Offline Dataset & Meta-training / Target Tasks.** We leverage a dataset of 600 human-teleoperated manipulation sequences of Gupta et al. (2019) for offline pre-training. In each trajectory, the robot executes a sequence of four subtasks. We then define a set of 23 meta-training tasks and 10 target tasks that in turn require the consecutive execution of four subtasks (see Figure 3 for examples). Note that each task consists of a unique combination of subtasks. More details can be found in Section G.2.

### 5.2 BASELINES

We compare SiMPL to prior approaches in RL, skill-based RL, meta-RL, and multi-task RL.

- **SAC** (Haarnoja et al., 2018) is a state of the art deep RL algorithm. It learns to solve a target task from scratch without leveraging the offline dataset nor the meta-training tasks.
- **SPiRL** (Pertsch et al., 2020) is a method designed to leverage offline data through the transfer of learned skills. It acquires skills and a skill prior from the offline dataset but does not utilize the meta-training tasks. This investigates the benefits our method can obtain from leveraging the meta-training tasks.
- **PEARL** (Rakelly et al., 2019) is a state of the art off-policy meta-RL algorithm that learns a policy which can quickly adapt to unseen test tasks. It learns from the meta-training tasks but does not use the offline dataset. This examines the benefits of using learned skills in meta-RL.
- **PEARL-ft** demonstrates the performance of a PEARL (Rakelly et al., 2019) model further fine-tuned on a target task using SAC (Haarnoja et al., 2018).
- **Multi-task RL (MTRL)** is a multi-task RL baseline which learns from the meta-training tasks by distilling individual policies specialized in each task into a shared policy, similar to Distral (Teh et al., 2017). Each individual policy is trained using SPiRL by leveraging skills extracted from the offline dataset. Therefore, it utilizes both the meta-training tasks and offline dataset similar to our method. This provides a direct comparison of multi-task learning (MTRL) from the training tasks vs. meta-learning using them (ours).

More implementation details on the baselines can be found in Section F.

### 5.3 RESULTS

We present the quantitative results in Figure 4 and the qualitative results on the maze navigation domain in Figure 5. In Figure 4, SiMPL demonstrates much better sample efficiency for learning the unseen target tasks compared to all the baselines. Without leveraging the offline dataset and meta-training tasks, SAC is not able to make learning progress on most of the target tasks. While PEARL is first trained on the meta-training tasks, it still achieves poor performance on the target tasks and fine-tuning it (PEARL-ft) does not yield significant improvement. We believe this is because both environments provide only sparse rewards yet require the model to exhibit long-horizon and complex behaviors, which is known to be difficult for meta-RL methods (Mitchell et al., 2021).

On the other hand, by first extracting skills and acquiring a skill prior from the offline dataset, SPiRL's performance consistently improves with more samples from the target tasks. Yet, it requires significantly more environment interactions than our method to solve the target tasks since the policy is optimized using vanilla RL, which is not designed to learn to quickly learn new tasks. While the multi-task RL (MTRL) baseline first learns a multi-task policy from the meta-training tasks, its sample efficiency is similar to SPiRL on target task learning, which highlights the strength of our proposed method – meta-learning from the meta-training tasks for fast target task learning.

Compared to the baselines, our method learns the target tasks much quicker. Within only a few episodes the policy converges to solve more than $80\%$ of the target tasks in the maze environment and

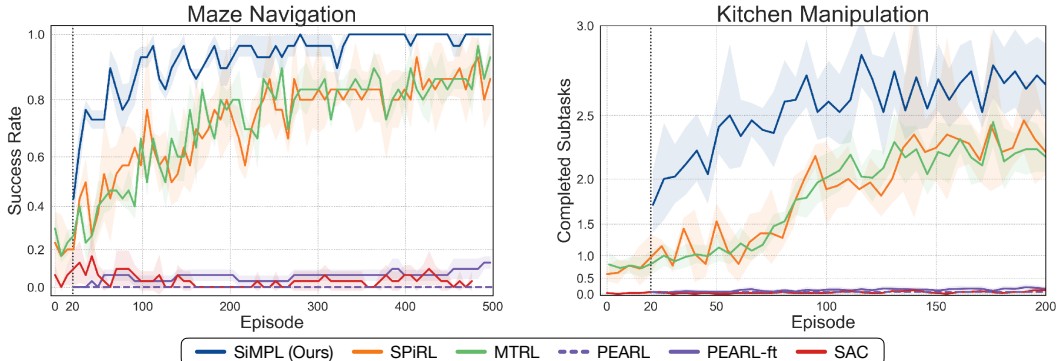

Figure 4: **Target Task Learning Efficiency.** SiMPL demonstrates better sample efficiency compared to all the baselines, verifying the efficacy of meta-learning on long-horizon tasks by leveraging skills and skill prior extracted from an offline dataset. For both the two environments, we train each model on each target task with 3 different random seeds. SiMPL and PEARL-ft first collect 20 episodes of environment interactions (vertical dotted line) for conditioning the meta-trained policy before fine-tuning it on target tasks.

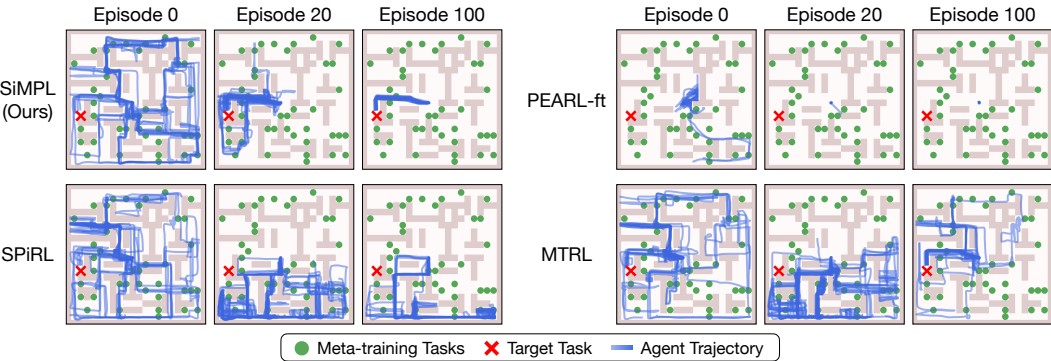

Figure 5: **Qualitative Results.** All the methods that leverage the offline dataset (*i.e.* SiMPL, SPiRL, and MTRL) effectively explore the maze in the first episode. Then, SiMPL converges with much fewer episodes compared to SPiRL and MTRL. In contrast, PEARL-ft is not able to make learning progress.

two out of four subtasks in the kitchen manipulation environment. The prior-regularized fine-tuning then continues to improve performance. The rapidly increasing performance and the overall faster convergence show the benefits of leveraging meta-training tasks in addition to learning from offline data: by first learning to learn how to quickly solve tasks using the extracted skills and the skill prior, our policy can efficiently solve the target tasks.

The qualitative results presented in Figure 5 show that all the methods that leverage the offline dataset (*i.e.* SiMPL, SPiRL, and MTRL) effectively explore the maze in the first episode. Then, SiMPL converges with much fewer episodes compared to SPiRL and MTRL, underlining the effectiveness of meta-training. In contrast, PEARL-ft is not able to make learning progress, justifying the necessity of employing offline datasets for acquiring long-horizon, complex behaviors.

## 5.4 META-TRAINING TASK DISTRIBUTION ANALYSIS

In this section, we aim to investigate the effect of the meta-training task distribution on our skill-based meta-training and target task learning phases. Specifically, we examine the effect of (1) the number of tasks in the meta-training task distribution and (2) the alignment between a meta-training task distribution and target task distribution. We conduct experiments and analyses in the maze navigation domain. More details on task distributions can be found in Section G.1.

**Number of meta-training tasks.** To investigate how the number of meta-training tasks affects the performance of our method, we train our method with fewer numbers meta-training tasks (*i.e.* 10 and 20) and evaluate it with the same set of target tasks. The quantitative results presented in Figure 6(a)

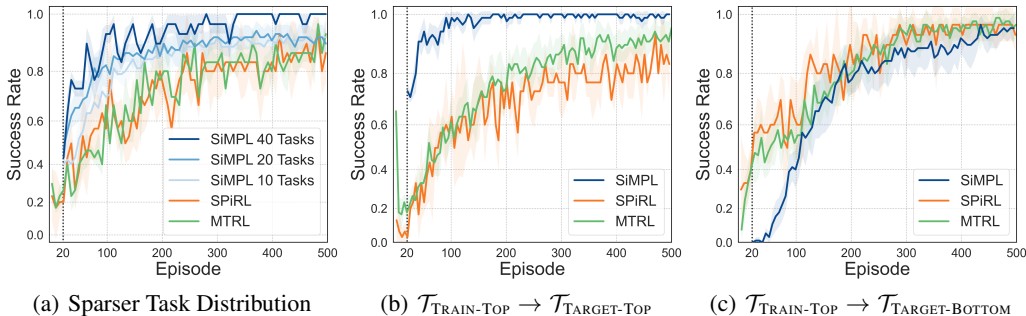

(a) Sparser Task Distribution    (b) $\mathcal{T}_{\text{TRAIN-TOP}} \rightarrow \mathcal{T}_{\text{TARGET-TOP}}$    (c) $\mathcal{T}_{\text{TRAIN-TOP}} \rightarrow \mathcal{T}_{\text{TARGET-BOTTOM}}$

Figure 6: **Meta-training Task Distribution Analysis.** (a) With sparser meta-training task distributions (*i.e.* fewer numbers of meta-training tasks), SiMPL still achieves better sample efficiency compared to SPiRL, highlighting the benefit of leveraging meta-training tasks. (b) When trained on a meta-training task distribution that aligns better with the target task distribution, SiMPL achieves improved performance. (c) When trained on a meta-training task distribution that is mis-aligned with the target tasks, SiMPL yields worse performance. For all the analyses, we train each model on each target task with 3 different random seeds.

suggest that even with sparser meta-training task distributions (*i.e.* fewer numbers of meta-training tasks), SiMPL is still more sample efficient compared to the best-performing baseline (*i.e.* SPiRL).

**Meta-train / test task alignment.** We aim to examine if a model trained on a meta-training task distribution that aligns better/worse with the target tasks would yield improved/deteriorated performance. To this end, we create biased meta-training / test task distributions: we create a meta-train set by sampling goal locations from only the top 25% portion of the maze ($\mathcal{T}_{\text{TRAIN-TOP}}$). To rule out the effect of the density of the task distribution, we sample 10 (*i.e.* $40 \times 25\%$) meta-training tasks. Then, we create two target task distributions that have good and bad alignment with this meta-training distribution respectively by sampling 10 target tasks from the top 25% portion of the maze ($\mathcal{T}_{\text{TARGET-TOP}}$) and 10 target tasks from the bottom 25% portion of the maze ($\mathcal{T}_{\text{TARGET-BOTTOM}}$).

Figure 6(b) and Figure 6(c) present the target task learning efficiency for models trained with good task alignment (meta-train on $\mathcal{T}_{\text{TRAIN-TOP}}$, learn target tasks from $\mathcal{T}_{\text{TARGET-TOP}}$) and bad task alignment (meta-train on $\mathcal{T}_{\text{TRAIN-TOP}}$, learn target tasks from $\mathcal{T}_{\text{TARGET-BOTTOM}}$), respectively. The results demonstrate that SiMPL can achieve improved performance when trained on a better aligned meta-training task distribution. On the other hand, not surprisingly, SiMPL and MTRL perform slightly worse compared to SPiRL when trained with misaligned meta-training tasks (see Figure 6(c)). This is expected given that SPiRL does not learn from the misaligned meta-training tasks. In summary, from Figure 6, we can conclude that meta-learning from either a diverse task distribution or a better informed task distribution can yield improved performance for our method.

## 6 CONCLUSION

We propose a skill-based meta-RL method, dubbed SiMPL, that can meta-learn on long-horizon tasks by leveraging prior experience in the form of large offline datasets without additional reward and task annotations. Specifically, our method first learns to extracts reusable skills and a skill prior from the offline data. Then, we propose to meta-trains a high-level policy that leverages these skills for efficient learning of unseen target tasks. To effectively utilize learned skills, the high-level policy is regularized by the acquired prior. The experimental results on challenging continuous control long-horizon navigation and manipulation tasks with sparse rewards demonstrate that our method outperforms the prior approaches in deep RL, skill-based RL, meta-RL, and multi-task RL. In the future, we aim to demonstrate the scalability of our method to high-DoF continuous control problems on real robotic systems to show the benefits of our improved sample efficiency.

## ACKNOWLEDGMENTS

This work was supported by the Engineering Research Center Program through the National Research Foundation of Korea (NRF) funded by the Korean Government MSIT (NRF-2018R1A5A1059921), KAIST-NAVER Hypercreative AI Center, and Institute of Information & communications Technology Planning & Evaluation (IITP) grant funded by the Korea government (MSIT) (No.2019-0-00075, Artificial Intelligence Graduate School Program (KAIST)). The authors are grateful for the fruitful discussion with the members of USC CLVR lab.

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

APPENDIX

# Table of Contents

## A  META-REINFORCEMENT LEARNING METHOD ABLATION

In this section, we compare the learning efficiency of different meta-RL algorithms with respect to the length of the training tasks. Specifically, we hypothesize that our approach SiMPL, which extracts temporally extended skills from offline experience, is better suited for learning long-horizon tasks than prior meta-RL algorithms. To cleanly investigate the importance of the temporally extended skills vs. the importance of using prior experience we include two additional comparisons to methods that leverage prior experience for meta-RL but via flat behavioral cloning instead of through temporally extended skills:

- **BC+PEARL** first learns a behavior cloning (BC) policy through supervised learning from the offline dataset. Then, analogous to our approach SiMPL, during the meta-training phase, a task encoder and a meta-learned policy are meta-trained with the BC policy constrained SAC objective. For fair comparison, we use the same residual policy parameterization as described in Section E.1.3.
- **BC+MAML** follows the same learning procedure described above, but uses MAML (Finn et al., 2017) for meta-training instead of PEARL. We follow the original learning objective in Finn et al. (2017) (*i.e.* using REINFORCE (Williams, 1992) for task adaptation, and using TRPO (Schulman et al., 2017) for meta-policy optimization).

We compare these methods as well as the standard meta-RL approach PEARL (Rakelly et al., 2019) on three meta-training tasks distributions of increasing complexity in the maze navigation environment (see Figure 7): (1) short-range goals with small variance $\mathcal{T}_{\text{TRAIN-EASY}}$, (2) short-range goals with larger variance $\mathcal{T}_{\text{TRAIN-MEDIUM}}$, and (3) long-range goals with large variance $\mathcal{T}_{\text{TRAIN-HARD}}$, which we used in our original maze experiments. By increasing variance and length of the tasks in each task distribution, we can investigate the learning capability of the meta-RL algorithms.

We present the quantitative results in Figure 8 and the corresponding qualitative analysis in Figure 9. On the simplest task distribution we find that all approaches can learn to solve the tasks efficiently, except for BC+MAML. While the latter also learns to solve the task eventually (see performance

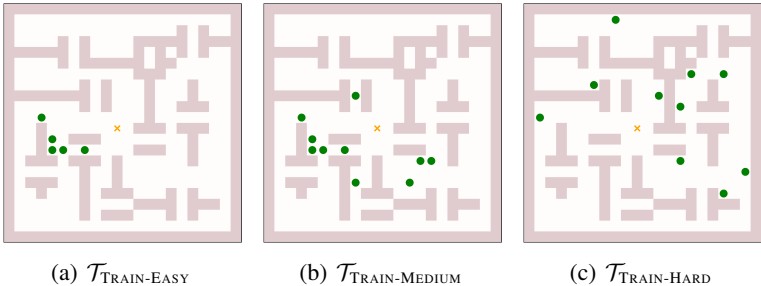

(a) $\mathcal{T}_{\text{TRAIN-EASY}}$       (b) $\mathcal{T}_{\text{TRAIN-MEDIUM}}$       (c) $\mathcal{T}_{\text{TRAIN-HARD}}$

Figure 7: **Task Distributions for Task Length Ablation.** We propose three meta-training task distributions of increasing difficulty to compare different meta-RL algorithms: $\mathcal{T}_{\text{TRAIN-EASY}}$ uses short-horizon tasks with adjacent goal locations, making exploration easier during meta-training, $\mathcal{T}_{\text{TRAIN-MEDIUM}}$ uses similar task horizon but increases the goal position variance, $\mathcal{T}_{\text{TRAIN-HARD}}$ contains long-horizon tasks with high variance in goal position and thus is the hardest of the tested task distributions.

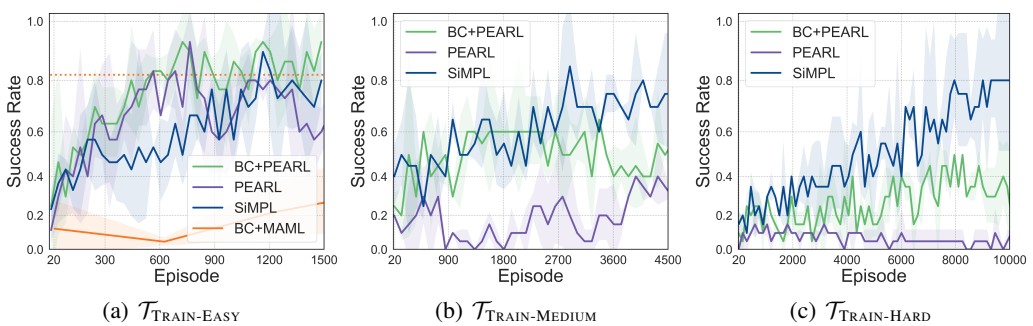

(a) $\mathcal{T}_{\text{TRAIN-EASY}}$       (b) $\mathcal{T}_{\text{TRAIN-MEDIUM}}$       (c) $\mathcal{T}_{\text{TRAIN-HARD}}$

Figure 8: **Meta-Training Performance for Task Length Ablation.** We find that most meta-learning approaches can solve the simplest task distribution, but using prior experience in BC+PEARL and SiMPL helps for the more challenging distributions (b) and (c). We find that only our approach, which uses the prior data by extracting temporally extended skills, is able to learn the challenging long-horizon tasks efficiently.

upon convergence as dashed orange line in Figure 8(a)) it uses on-policy meta-RL and thus requires substantially more environment interactions during meta-training. We thus only consider the more sample efficient BC+PEARL off-policy meta-RL method in the remaining comparisons.

On the more complex task distributions $\mathcal{T}_{\text{TRAIN-MEDIUM}}$ and $\mathcal{T}_{\text{TRAIN-HARD}}$, we find that using prior data for meta-learning is generally beneficial: both BC+PEARL and SiMPL learn more efficiently on the task distribution of medium difficulty $\mathcal{T}_{\text{TRAIN-MEDIUM}}$, as shown in Figure 8(b), since the policy pre-trained from offline data allows for more efficient exploration during meta-training. Importantly, on the hardest task distribution $\mathcal{T}_{\text{TRAIN-HARD}}$, as shown in Figure 8(c), which consists exclusively of long-horizon tasks, we find that only SiMPL is able to effectively learn, highlighting the importance of leveraging the offline data via temporally extended skills instead of flat behavioral cloning. This supports our intuition that the abstraction provided by skills is particularly beneficial for meta-learning on long-horizon tasks.

## B    LEARNING EFFICIENCY ON TARGET TASKS WITH FEW EPISODES OF EXPERIENCE

In this section, we examine the data efficiency of the compared methods on the target tasks, specifically when provided with only a *few* (<20) episodes of online interaction with an unseen target task. Being able to learn new tasks this quickly is a major strength of meta-RL approaches (Finn et al., 2017; Rakelly et al., 2019). We hypothesize that our skill-based meta-RL algorithm SiMPL can learn similarly fast, even on long-horizon, sparse-reward tasks.

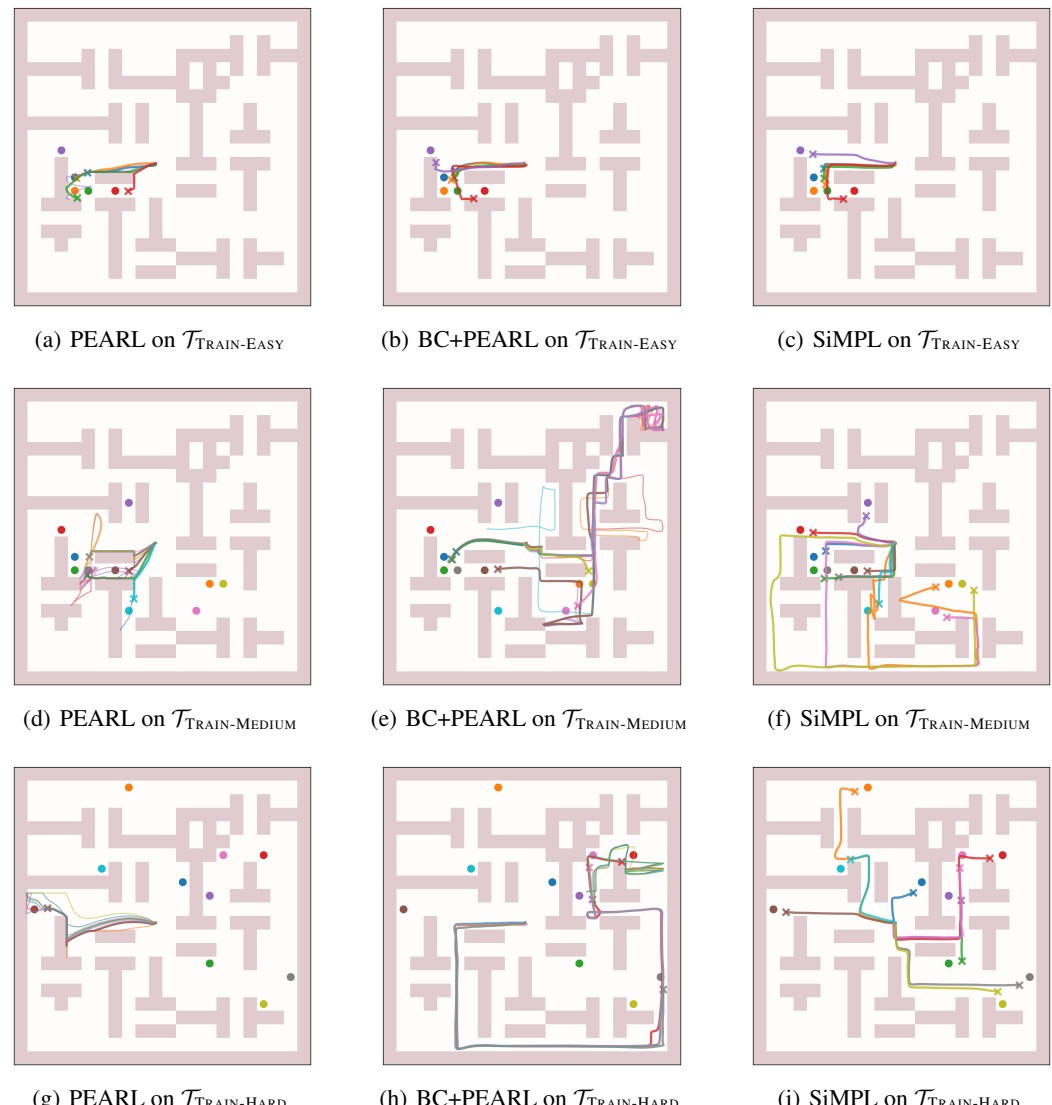

Figure 9: **Qualitative Result of Meta-reinforcement Learning Method Ablation. Top.** All the methods can learn to solve short-horizon tasks $\mathcal{T}_{\text{TRAIN-EASY}}$. **Middle.** On medium-horizon tasks $\mathcal{T}_{\text{TRAIN-MEDIUM}}$, PEARL struggles at exploring further, while BC+PEARL exhibits more consistent exploration yet still fails to solve some of the tasks. SiMPL can explore well and solve all the tasks. **Bottom.** On long-horizon tasks $\mathcal{T}_{\text{TRAIN-HARD}}$, PEARL falls into a local minimum, focusing only on one single task on the left. BC+PEARL explores slightly better and can solve a few more tasks. SiMPL can effectively learn all the tasks.

In our original evaluations in Section 5, we used 20 episodes of initial exploration to condition our meta-trained policy. In Figure 10, we instead compare performance of different approaches when only provided with very few episodes of online interactions. We find that SiMPL learns to solve the unseen tasks substantially faster than all alternative approaches. On the kitchen manipulation tasks our approach learns to almost solve two out of four subtasks within a time span equivalent to only a few minutes of real-robot execution time. In contrast, prior meta-RL methods struggle at making progress at all on such long-horizon tasks, showing the benefit of combining meta-RL with prior offline experience.

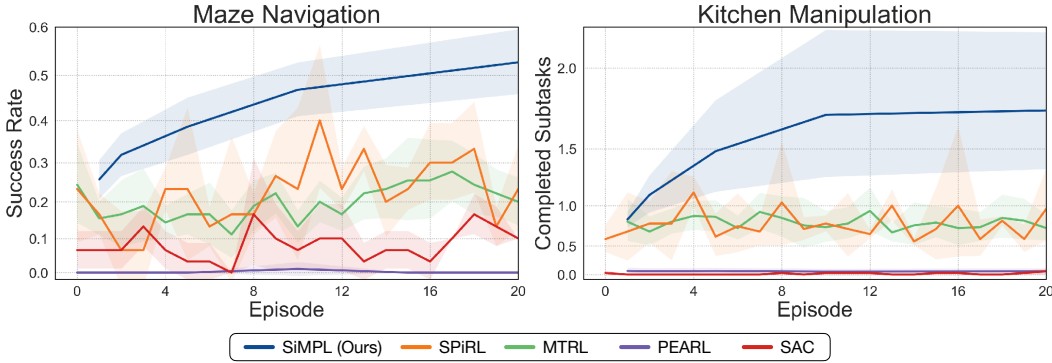

Figure 10: **Performance with few episodes of target task interaction.** We find that our skill-based meta-RL approach SiMPL is able to learn complex, long-horizon tasks within few episodes of online interaction with a new task while prior meta-RL approaches and non-meta-learning baselines require many more interactions or fail to learn the task altogether.

## C    INVESTIGATING OFFLINE DATA VS. TARGET DOMAIN SHIFT

To provide more insights on comparing SiMPL and SPiRL (Pertsch et al., 2020), we evaluate SiMPL in the maze navigation task setup proposed in Pertsch et al. (2020). This tests whether our approach can scale to image-based observations: Pertsch et al. (2020) use $32 \times 32$px observations centered around the agent. Even more importantly, it allows us to investigate the robustness of the approach to the domain shifts between the offline pre-training data and the target task: we use the maze navigation offline dataset from Pertsch et al. (2020) which was collected on *randomly sampled* $20 \times 20$ maze layouts and test on tasks in the unseen, randomly sampled $40 \times 40$ test maze layout from Pertsch et al. (2020). We visualize the meta-training task distribution in Figure 11(a) and the target task distribution in Figure 11(b).

We compare the performance of our method to the best-performing baseline, SPiRL (Pertsch et al., 2020), in Figure 11(c). Similar to the result presented in Figure 4, SiMPL can learn the target task faster by combining skills learned from the offline dataset with efficient meta-training. This shows that our approach can scale to image-based inputs and is robust to substantial domain shifts between the offline pre-training data and the target tasks.

Note that the above results are obtained by comparing our proposed method and SPiRL with the exact same setup used in the SPiRL paper (Pertsch et al., 2020). Specifically, we used the same initial position of the agent as well as sampled the tasks of comparable complexity to the ones used in the SPiRL paper for our evaluation (please see Figure 13 in the SPiRL paper for tasks used in their evaluation). While the used test tasks do not fully cover the entire maze, they are already considerably long-horizon, requiring on average 710 steps until completion while only providing sparse goal-reaching rewards.

To further explore the performance of our proposed method and SPiRL, we have experimented with learning from goals sampled across the entire maze. Yet, SPiRL cannot learn such target tasks and our proposed method consequently does not converge well on the meta-training tasks. This highlights the limitation of skill-based RL methods and can potentially be addressed by learning a more expressive skill prior, *e.g.* using flow models (Dinh et al., 2017), but this is outside the scope of this work.

## D    EXTENDED RELATED WORK

We present an extended discussion of the related work in this section.

**Pre-training in Meta-learning.** Leveraging pre-trained models for improving meta-learning methods has been explored in Bronskill et al. (2021); Dvornik et al. (2020); Kolesnikov et al. (2020); Triantafillou et al. (2020) with a focus on few-shot image classification. One can also view our proposed framework as a meta-reinforcement learning method with a pre-training phase. Specifically, in the pre-training phase, we propose to first extract reusable skills and a skill prior from offline

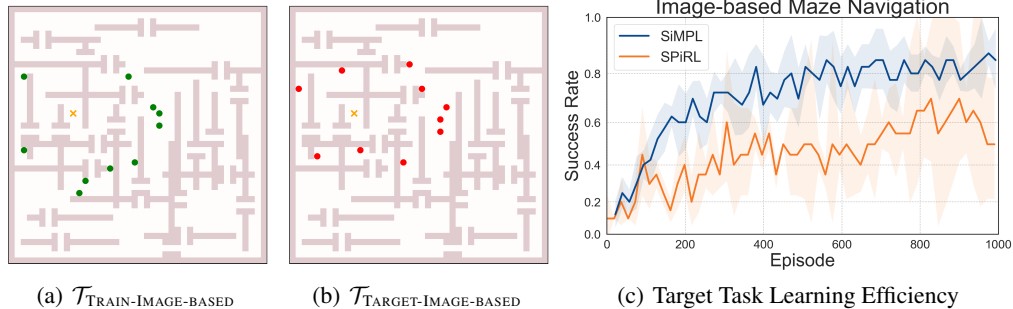

(a) $\mathcal{T}_{\text{TRAIN-IMAGE-BASED}}$     (b) $\mathcal{T}_{\text{TARGET-IMAGE-BASED}}$     (c) Target Task Learning Efficiency

Figure 11: **Image-Based Maze Navigation with Distribution Shift.** **(a-b)**: Meta-training and target task distributions. The green dots represent the goal locations of meta-training tasks and the red dots represent the goal locations of target tasks. The yellow cross represent the initial location of the agent, which is equivalent to the one used in Pertsch et al. (2020). **(c)**: Performance on the target task. Our approach SiMPL can leverage skills learned from offline data for efficient meta-RL on the maze navigation task and is robust to the domain shift between offline data environments and the target environment.

datasets without reward or task annotations in a self-supervised fashion. Then, our proposed method meta-learns from a set of meta-training tasks, which significantly accelerates learning on unseen target tasks.

# E   IMPLEMENTATION DETAILS ON OUR METHOD

In this section, we describe the additional implementation details on our proposed method. The details on model architecture is presented in Section E.1, followed by the training detailed described in Section E.2.

## E.1   MODEL ARCHITECTURE

We describe the details on our model architecture in this section.

### E.1.1   SKILL PRIOR

We followed architecture and learning procedure of Pertsch et al. (2020) for learning a low-level skill policy and a skill prior. Please refer to Pertsch et al. (2020) for more details on the architectures for learning skills and skill priors from offline datasets.

### E.1.2   TASK ENCODER

Following Rakelly et al. (2019), our task encoder is a permutation invariant neural network. Specifically, we adopt Set Transformer (Lee et al., 2019) that consists of layers $[2 \times \text{ISAB}_{32} \to \text{PMA}_1 \to 3 \times \text{MLP}]$ for expressive and efficient set encoding. All the hidden layers are 128-dimensional and all attention layers have 4 attention heads. The encoder takes a set of high-level transitions as input, where each transition is a vector concatenation of high-level transition tuple. The output of the encoder is $(\mu_e, \sigma_e)$ which are the parameters of Gaussian task posterior $p(e|c) = \mathcal{N}(e; \mu_e, \sigma_e)$. We varied task vector dimension $\dim(e)$ depends on task distribution complexity. $\dim(e) = 10$ for Kitchen Manipulation, $\dim(e) = 6$ for Maze Navigation with 40 meta-training tasks, and $\dim(e) = 5$, otherwise.

### E.1.3   POLICY

We parameterize our policy with neural network. We employed 4-layer MLPs with 256 hidden units for Maze Navigation, and 6-layer MLPs with 128 hidden unit for Kitchen Manipulation experiment. Instead of direct parameterization of policy, the network output is added to skill-prior to make learning more stable. Specifically, the policy network takes concatenation of $(s, e)$ as input, and then

outputs residual parameters $(\mu_z, \log \sigma_z)$ to skill-prior distribution $p(z|s) = \mathcal{N}(z|\mu_p, \sigma_p)$. Resulting distribution by this residual parameterization is $\pi(z|s) = \mathcal{N}(z|\mu_p + \mu_z, \exp(\log \sigma_p + \log \sigma_z))$

### E.1.4 CRITIC

The critic network takes concatenation of $s$, $e$, and skill $z$ as input and outputs an estimation of task-conditioned Q-value $Q(s, z, e)$. We employ double Q networks (Van Hasselt et al., 2016) to mitigate Q-value overestimation. The architecture of critic follows the policy.

### E.2 TRAINING DETAILS

For all the network updates, we used Adam optimizer (Kingma & Ba, 2015) with a learning rate of $3e - 4$, $\beta_1 = 0.9$, and $\beta_2 = 0.999$. We describe the training details of the skill-based meta-training phase in Section E.2.1 and the target task learning phase Section E.2.2.

### E.2.1 SKILL-BASED META-TRAINING

Our meta-training procedure is similar to the procedure adopted in (Rakelly et al., 2019). Encoder and critics networks are updated to minimize MSE between Q-value prediction and target Q value. Policy network is updated to optimize Equation 4 without updating the encoder network. All network are updated with the average gradients of 20 randomly sampled tasks. Each batch of gradients is computed from 1024 and 256 transitions for Maze Navigation and Kitchen Manipulation experiment, respectively. We train our models for 10000, 18000, and 16000 episodes for the Maze Navigation experiments with 10, 20, 40 meta-training tasks, respectively, and 3450 episodes for Kitchen Manipulation.

As stated in Section 4.2, we apply different regularization coefficients depending on the size of the conditioning transitions. In Maze Navigation experiment, we set target KL divergence to $0.1$ for batch that is conditioned on size 4 transitions and $0.4$ for batch conditioned on size 8192 transitions. In Kitchen Manipulation experiment, we set target KL divergence to $0.4$ for batch conditioned with a size 1024 transitions while KL coefficient for batch conditioned on size 2 transitions is fixed to $0.3$.

### E.2.2 TARGET TASK LEARNING

We initialize the Q function and the auto-tuning value $\alpha$ with the values that learned in the skill-based meta-training phase. The policy is initialized after observing and encoding 20 episodes obtained from the task unconditioned policy rollouts. For the target task learning phase, the target KL $\delta$ is 1 for Maze Navigation, and 2 for Kitchen Manipulation experiments. To compute a gradient step, 256 high-level transitions are sampled from a replay buffer with size 20000. Note that we used same setup for baselines that uses SPiRL fine-tuning (SPiRL and MTRL).

## F IMPLEMENTATION DETAILS ON BASELINES

In this section, we describe the additional implementation details on producing the results of the baselines.

### F.1 SAC

The SAC (Haarnoja et al., 2018) baseline learns to solve a target task from scratch without leveraging the offline dataset nor the meta-training tasks.

We initialize $\alpha$ to 0.1 and set the target entropy to $\mathcal{H} = -\dim(\mathcal{A})$. To compute a gradient step, 4096 and 1024 environment transitions are sampled from a replay buffer for Maze Navigation and Kitchen Navigation experiments, respectively.

### F.2 PEARL AND PEARL-FT

PEARL (Rakelly et al., 2019) learns from the meta-training tasks but does not use the offline dataset. Therefore, we directly train PEARL models on the meta-training tasks without the phase of learning

from offline datasets. We use gradients averaged from 20 randomly sampled tasks where each task gradient is computed by batch sampled from a per-task buffer. The target entropy is set to $\mathcal{H} = -\dim(\mathcal{A})$ and $\alpha$ is initialized to $0.1$.

While the method proposed in Rakelly et al. (2019) does not fine-tune on target/meta-testing tasks, we extend PEARL to be fine-tuned on target tasks for a fair comparison, called PEARL-ft. Since PEARL does not use learned skills or a skill prior, the target task learning of PEARL is simply running SAC with task-encoded initialization. Similar to the target task learning of our method, we initialize the Q function and entropy coefficient $\alpha$ to the value learned during the meta-training phase. Also, we initialize the policy to the task conditioned policy after observing 20 episodes of experience from the task unconditioned policy rollouts. The hyperparameters used for fine-tuning are the same as SAC.

### F.3 SPiRL

Similar to our method, we initialize the high-level policy to skill-prior while fixing low-level policy for target task learning for SPiRL. $\alpha$ is initialized to $0.01$ and we use the same hyperparameters for the SPiRL models as our method.

### F.4 MULTI-TASK RL (MTRL)

Inspired by Distral (Teh et al., 2017), our multi-task RL baseline is designed to first learns a set of individual policies, where each of them is specialized in one task; then, a shared/multi-task policy is learned by distilling the individual polices. Since it is inefficient to learn an individual policy from scratch, we learn each individual policy using SPiRL with learned skills and a skill prior. Then, we distill the individual policies using the following objective :

$$\max_{\pi_0} \mathbb{E}_{\mathcal{T} \sim p_{\mathcal{T}}} \left[ \sum_t \mathbb{E}_{(s_t, z_t) \sim \rho_{\pi_0}} \left[ r_{\mathcal{T}}(s_t, z_t) - \alpha D_{\mathrm{KL}}\big( \pi_0(z|s_t, e), p(z|s_t) \big) \right] \right]. \tag{6}$$

We use the same setup for $\alpha$ as our method, where $\alpha$ is auto-tuned to satisfy a target KL, $\delta = 0.1$ for Maze Navigation and $\delta = 0.2$ for Kitchen Manipulation.

While the target task learning phase for MTRL is similar to ours, except that MTRL is not initialized with a meta-trained Q function and learned $\alpha$.

## G META-TRAINING TASKS AND TARGET TASKS.

In this section, we present the meta-training tasks and target tasks used in the maze navigation domain and the kitchen manipulation domain.

### G.1 MAZE NAVIGATION

The meta-training tasks and target tasks are visualized in Figure 12 and Figure 13.

### G.2 KITCHEN MANIPULATION

The meta-training tasks are:

- microwave→kettle→bottom burner→slide cabinet
- microwave→bottom burner→top burner→slide cabinet
- microwave→top burner→light switch→hinge cabinet
- kettle→bottom burner→light switch→hinge cabinet
- microwave→bottom burner→hinge cabinet→top burner
- kettle→top burner→light switch→slide cabinet
- microwave→kettle→slide cabinet→bottom burner
- kettle→light switch→slide cabinet→bottom burner

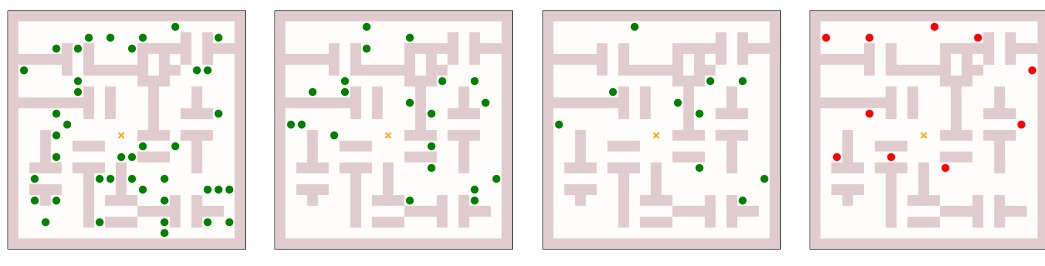

(a) Meta-training 40 Tasks    (b) Meta-training 20 Tasks    (c) Meta-training 10 Tasks      (d) Target Tasks

Figure 12: **Maze Meta-training and Target Task Distributions.** The green dots represent the goal locations of meta-training tasks and the red dots represent the goal locations of target tasks. The yellow cross represent the initial location of the agent.

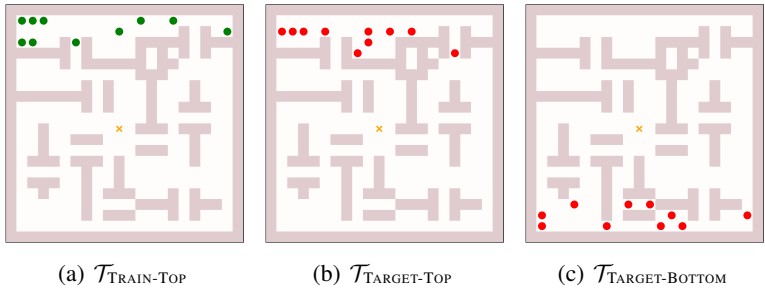

(a) $\mathcal{T}_{\text{TRAIN-TOP}}$        (b) $\mathcal{T}_{\text{TARGET-TOP}}$        (c) $\mathcal{T}_{\text{TARGET-BOTTOM}}$

Figure 13: **Maze Meta-training and Target Task Distributions for Meta-training Task Distribution Analysis.** The green dots represent the goal locations of meta-training tasks and the red dots represent the goal locations of target tasks. The yellow cross represent the initial location of the agent.

- microwave→kettle→bottom burner→top burner
- microwave→kettle→slide cabinet→hinge cabinet
- microwave→bottom burner→slide cabinet→top burner
- kettle→bottom burner→light switch→top burner
- microwave→kettle→top burner→light switch
- microwave→kettle→light switch→hinge cabinet
- microwave→bottom burner→light switch→slide cabinet
- kettle→bottom burner→top burner→light switch
- microwave→light switch→slide cabinet→hinge cabinet
- microwave→bottom burner→top burner→hinge cabinet
- kettle→bottom burner→slide cabinet→hinge cabinet
- bottom burner→top burner→slide cabinet→light switch
- microwave→kettle→light switch→slide cabinet
- kettle→bottom burner→top burner→hinge cabinet
- bottom burner→top burner→light switch→slide cabinet

The target tasks are:

- microwave→bottom burner→light switch→top burner
- microwave→bottom burner→top burner→light switch
- kettle→bottom burner→light switch→slide cabinet
- microwave→kettle→top burner→hinge cabinet
- kettle→bottom burner→slide cabinet→top burner

- kettle→light switch→slide cabinet→hinge cabinet
- kettle→bottom burner→top burner→slide cabinet
- microwave→bottom burner→slide cabinet→hinge cabinet
- bottom burner→top burner→slide cabinet→hinge cabinet
- microwave→kettle→bottom burner→hinge cabinet

