# OpenReview forum: "Skill-based Meta-Reinforcement Learning"
_ICLR.cc/2022/Conference — ICLR 2022 Poster_

### Official Review · Reviewer_1NZY · 2021-10-27

**Correctness:** 3
**Technical Novelty And Significance:** 2
**Empirical Novelty And Significance:** 3
**Recommendation:** 6
**Confidence:** 4

**Main Review:**

Strengths
----------

- The paper tackles the problem of learning over a long-horizon, which is an important challenge.
- The experiments show a clear improvement over methods like SPiRL and PEARL.
- Additional experiments on number of tasks and task misalignment are a valuable source of information.
- The paper is well written and easy to follow.

Weaknesses
----------

- Novelty and contribution. The method seems to be a variant of Rakelly et al. (2019) which uses the learnable priors from Pertsch et al. (2020). Overall, the difference with those methods is negligible. The major contribution of the paper is the use of a pre-training stage before meta-training which is a marginal contribution, since pre-training has been already used in meta-learning. I am open to change my mind on this point if the authors can provide concrete evidences of the relevance and novelty of their work.

- Baselines. It is missing a comparison against standard meta-training methods. The authors state that meta-agents cannot learn long-horizon behaviours. However, I did not see any convincing evidence to backup this claim in the environments considered in the paper. A basic experiment would be to pre-train a standard backbone using offline data (e.g. via behavioural cloning), and then use this backbone to train a classical meta-learner such as MAML (Finn et al. 2017), CAVIA (Zintgraf et al. 2019), or more recent variants. The backbone could be finetuned or frozen at meta-training time. Pre-training the backbone of the meta-learner will guarantee the same exposure to the training environment resulting in a fairer comparison with the other methods.

- Domain shift. The authors have explored the performance under misaligned train/test tasks (see Section 5.4, Figures 6b and 6c) but there is not much regarding misalignment between pre-train/meta-train. In Section 3 (Problem Formulation) the authors state that they do not assume that the offline data is part of the set of training tasks nor that it contains demonstrations for solving them. However, it seems to me that the offline data is collected from the same environment, enforcing a strong inductive bias on the pre-trained model. It is not clear if the method can still work under a mild/strong domain shift. In other words, it is not clear the type of constraints one should impose on the offline data in order to guarantee robust performances at test time. It would be beneficial to see an empirical comparison using different offline data to check how they affect the performance. Different sets of offline data could be produced via domain randomization over the base environment. This is an important factor to explore, since it could lead to a performance drop similarly to the one observed in the misaligned tasks experiments.

- Long-horizon. One of the main claim of the authors is that the method is able to cope with long-horizon. It would be interesting to quantify the performance of the method under environments with an increasing horizon. In particular, it would be useful to know when the method starts to be beneficial compared to standard meta-learner. In the maze environment this could be achieved by gradually increasing the distance between the starting point and the reward.

Post rebuttal
----------------

The authors have provided a new set of experiments that (partially) answer my concerns and I will raise my score to 6. Overall the paper is still short in terms of novelty and the new experiments have been carried on a simplistic environment. I have provided a more detailed feedback in the comments section.

References
----------

Finn, C., Abbeel, P., & Levine, S. (2017, July). Model-agnostic meta-learning for fast adaptation of deep networks. In International Conference on Machine Learning (pp. 1126-1135). PMLR.

Pertsch, K., Lee, Y., & Lim, J. J. (2020). Accelerating reinforcement learning with learned skill priors. arXiv preprint arXiv:2010.11944.

Rakelly, K., Zhou, A., Finn, C., Levine, S., & Quillen, D. (2019, May). Efficient off-policy meta-reinforcement learning via probabilistic context variables. In International conference on machine learning (pp. 5331-5340). PMLR.

Zintgraf, L., Shiarli, K., Kurin, V., Hofmann, K., & Whiteson, S. (2019, May). Fast context adaptation via meta-learning. In International Conference on Machine Learning (pp. 7693-7702). PMLR.

**Summary Of The Paper:**

In this paper the authors propose to learn long-horizon policies by following three stages: (i) pretrain a model on offline data to acquire a set of skills, (ii) meta-train a policy to use those skills, and (iii) adapt the meta-trained policy to solve an unseen target task. A series of experiments on maze navigation and manipulation tasks is provided to evaluate the method.

**Summary Of The Review:**

Given the marginal novelty and the lack of empirical evidences to backup the authors' claim I do not recommend to accept the paper. I am open to change my mind if the authors can provide a convincing rebuttal and more solid experiments.

---

> ### Author Response · Authors · 2021-11-23
> **Response to Reviewer 1NZY (2/2)**
>
> **Domain shift between offline data and meta-training tasks**: We appreciate the reviewer for this suggestion. To show that our approach can handle substantial environment shifts between the offline data collection and the meta-train tasks, we conduct an additional experiment where we leverage the offline dataset originally published with the SPiRL paper (Pertsch et al., 2020). This offline data is collected across a wide range of randomly sampled 20x20 maze layouts. Following SPiRL we then meta-train and test on a completely unseen 40x40 maze layout. In Section C, we show that our approach is still able to leverage skills learned from this data with substantial domain shift for successful meta-training and downstream task learning in the unseen test domain -- learning the task substantially faster than a SPiRL baseline. More generally, similar to prior works in skill-based RL, we need to assume that the offline data contains meaningful short-horizon behaviors which can be recombined to learn the target task behaviors. We explicitly do not require the offline experience to contain the long-horizon behaviors needed to solve the target task and can thus handle substantial domain shifts between offline data and meta-train/test tasks.
>
> **Performance under different task horizons**: We thank the reviewer for this suggestion. As proposed by the reviewer, we compare the performance of our approach to standard meta-RL methods (among others) on maze navigation tasks of varying length in Section A. We find that standard meta-learners are able to learn the tasks only on comparably short-horizon problems (see Figure 7, left — average task-length 170 steps) while only our approach is able to scale to the longer-horizon tasks in Figure 7, right (average task-length 280 steps). This clearly shows the benefit of learning temporally extended skills for more efficient exploration in long-horizon tasks during meta-training.
>
> Again, thank you for your insightful suggestions — we believe that the added experiments strengthen the claims of the paper. We hope that we were able to address all major concerns raised in your review. Please kindly let us know if there are any further concerns or missing experimental results that potentially prevent you from accepting this submission. We would be more than happy to address them. Thank you very much for all your detailed feedback and the time you put into helping us to improve our submission.

---

> > ### Comment · Reviewer_1NZY · 2021-11-24
> > **Answer to rebuttal**
> >
> > Thank you for your detailed answer. Overall you have addressed most of my concerns and I will increase my score. The paper is still short in terms of novelty and the experiments on Meta-RL baselines and horizon have only been carried on the maze environment, which is too simplistic. For these reasons I can only raise my score to 6.
> >
> > - **Novelty** The points you have highlighted show an improvement over previous work, with the main contribution being the use of additional techniques to stabilize training. However, as I said above I think that this contribution is still short in terms of novelty.
> >
> > - Regarding you question on the use of **pre-training in meta-learning**. While I am not aware of its usage in Meta-RL, this has become standard practice in state of the art meta-learning methods for classification. Most of the recent papers use pre-trained models, e.g. Bronskill et al. (2021), Dvornik et al. (2020), Kolesnikov et al. (2019), Triantafillou et al. (2019). I suggest to mention these references in the updated version of the paper, pointing out that while common in classification, pre-training in RL is not standard practice (as far as I know).
> >
> > - **BC + Meta-RL experiments** Those are useful baselines. It is interesting to see the boost in BC+PEARL and I am not surprised by the issues in BC+MAML which is unstable by nature. Comparing against more recent Meta-RL methods would be better.
> >
> > - **Domain shift and Task horizon** The additional experiments suggest that the method is robust to shift and that it can scale to longer horizons. However, the maze environment is too simplistic and experiments on more complex settings would be beneficial to better understand the nature of the problem.
> >
> > References
> > ---------------
> >
> > Bronskill, J. F., Massiceti, D., Patacchiola, M., Hofmann, K., Nowozin, S., & Turner, R. E. (2021, May). Memory Efficient Meta-Learning with Large Images. In Thirty-Fifth Conference on Neural Information Processing Systems.
> >
> > Dvornik, N., Schmid, C., & Mairal, J. (2020, August). Selecting relevant features from a multi-domain representation for few-shot classification. In European Conference on Computer Vision (pp. 769-786). Springer, Cham.
> >
> > Kolesnikov, A., Beyer, L., Zhai, X., Puigcerver, J., Yung, J., Gelly, S., & Houlsby, N. (2020). Big transfer (bit): General visual representation learning. In Computer Vision–ECCV 2020: 16th European Conference, Glasgow, UK, August 23–28, 2020, Proceedings, Part V 16 (pp. 491-507). Springer International Publishing.
> >
> > Triantafillou, E., Zhu, T., Dumoulin, V., Lamblin, P., Evci, U., Xu, K., ... & Larochelle, H. (2019). Meta-dataset: A dataset of datasets for learning to learn from few examples. arXiv preprint arXiv:1903.03096.

---

> > > ### Author Response · Authors · 2021-11-24
> > > **Re: Answer to rebuttal**
> > >
> > > We thank the reviewer for acknowledging our rebuttal and additional experiments. Please find the response to your further questions below.
> > >
> > > **Novelty and pre-training in meta-learning**:
> > > As pointed out by the reviewers, while pre-training in meta-learning has been widely employed in classification, it is not a standard practice in meta-RL. In our work, we propose to leverage offline datasets by extracting skills and skill prior, which can be viewed as “pre-training”. This “pre-training” then enables further meta-learning on long-horizon, sparse-reward tasks, which existing meta-RL methods struggle at.
> > > We believe that introducing this “pre-training”/skill extraction phase, proposing a way to implement this idea, and incorporating this with meta-learning is how our work is novel.
> > >
> > > We thank the reviewer for providing these references on meta-learning in classification. Since it is not possible to submit a rebuttal revision anymore, we will include these references in the related work upon acceptance. Also, we will revise the paper to discuss this perspective of pre-training in meta-RL suggested by the reviewer.
> > >
> > > **BC + Meta-RL experiments**:
> > > We thank the reviewer for recognizing the insights provided by our BC + Meta-RL experiments.
> > >
> > > **Domain shift and task horizon**:
> > > We completely agree with the reviewer and our immediate future work is to demonstrate the scalability of our method to more complex continuous control problems. Based on the consistent results across the maze environment and the kitchen environment (Figure 4), as well as the maze task distributions with different levels of difficulty (Section A), we are confident that the performance gain of our proposed method would translate well to more complex setups.
> > >
> > > We hope that this addresses your concern and makes the novelty of our work clear. Please kindly let us know if there are any further questions or concerns. We would be more than happy to address them.

---

> ### Author Response · Authors · 2021-11-23
> **Response to Reviewer 1NZY (1/2)**
>
> We thank the reviewer for the thorough and constructive comments. Please find the response to your questions below.
>
> **Novelty**: The main contribution of our work is the usage of task-agnostic experience to enable meta-learning on sparse, long-horizon tasks. We would like to emphasize that while our method builds on prior work in skill-based RL and meta-learning, our proposed method involves several novel techniques that are essential for enabling stable meta-learning with learned skills:
>
> - **Residual policy learning (see Section D.1.3)**: we parameterize our policy to predict the residual, i.e. the difference, with respect to the pre-trained skill prior instead of directly outputting the skill distribution. The residual parametrization stabilizes learning by ensuring that the initial skill distribution is close to the skill prior (note that we cannot simply initialize the policy to the skill prior, as done in Pertsch et al., 2020, since we additionally need to condition the policy on the context vector c).
> - **Adaptive policy regularization (see Section 4.2)**: we train our approach with variable sizes of conditioning transition sets. Since the amount of task information that is contained in this set can vary widely depending on the set’s size, particularly in sparse reward tasks, our method uses varying target KL divergence values for regularizing the policy towards the prior (stronger regularization for smaller transition sets), while prior works apply regularization with constant weight (Rakelly et al., 2019).
> - **Transfer of meta-trained Q-function (see Section D.2.2)**: together with the meta-trained policy, the Q-function and KL regularization coefficient are also transferred to the target task learning phase. This warm start of the Q-function prevents the policy from instantly falling back to the skill prior, which happens when using a randomly initialized Q-function and $alpha_0$ = 1 as in prior work (Pertsch et al., 2020).
>
> Without these techniques, we were unable to reliably meta-train skill-based policies, especially on the harder kitchen environment tasks. We believe that these techniques have not been employed in the aforementioned prior works and thus, together with the problem setting, constitute the novelty and technical contributions of our work. While the original submission already detailed these techniques, we have now revised the paper to underline their importance for stable meta-training in Section 4 and detail them in Section D.
>
> Question: the review mentions prior works that use pre-training in meta-learning — could the reviewer please provide some concrete references so that we can properly discuss them in the related work section? Thank you.
>
> **BC + Meta-RL methods**: We sincerely thank the reviewer for this insightful suggestion. As suggested by the reviewer, we added comparisons to approaches that first learn a behavioral cloning (BC) policy on the offline dataset and then leverage it for meta-training on the provided set of meta-training tasks. We show this evaluation in Section A (including both BC+MAML and BC+PEARL). Concretely, we conduct comparisons on three task distributions in the maze navigation domain with increasing difficulties: (1) short-range goals with small variance, (2) short-range goals with larger variance, and (3) long-range goals with a large variance which we used in our original maze experiments. The findings of this experiments are as follows:
> - **BC initialization helps**: BC initialization indeed helps meta-training as BC+PEARL outperforms PEARL. While vanilla PEARL can only solve the simplest scenario (1), the BC-initialized meta-learner (BC+PEARL) is able to explore better and can thus handle goals with larger variance  (i.e. solve scenarios (1) and (2)). This aligns with the reviewer's intuition.
> - **BC+PEARL outperforms BC+MAML**: comparing BC+MAML and BC+PEARL, we find that the former requires many more environment interactions to learn the target task since it is an on-policy meta-RL algorithm while BC+PEARL can leverage off-policy meta-RL updates and is thus substantially more sample efficient.
> - **BC+PEARL struggles at learning long-horizon tasks**: BC+PEARL still struggles at learning the longer-horizon tasks (3) since the BC policy does not learn temporally abstracted behaviors, i.e. skills. In contrast, we find that our approach can meta-learn on all the task distributions, which highlights the advantage of leveraging the offline experience through learned, temporally extended skills.
>
> We have revised the paper to include this experiment. For the detailed description of the experiment setup and the results, please refer to Section A.
>
> Additionally, we have included the reference (“Fast Context Adaptation via Meta-Learning” in ICML 2019 by Luisa M Zintgraf et al.) provided by the reviewer in the revised related work section.

---

### Official Review · Reviewer_jZvC · 2021-10-30

**Correctness:** 4
**Technical Novelty And Significance:** 3
**Empirical Novelty And Significance:** 3
**Recommendation:** 8
**Confidence:** 4

**Main Review:**

This work succeeds at combining the best of skill-based RL and meta-learning using task embeddings.  This combination introduces meta-learning at the skill level.  The results make a good case for the efficacy of the approach.

On the downside, the resulting system is quite a complicated beast.  While the ideas and methods are quite clear at a high level, the details are hard to follow, at least without deep familiarity with the Pertsch and Rakelly work.  Some of the notation is unclear, confusing or even wrong:
- In skill-based RL (p. 3), both policies are called "skill policies".  I'd call only the higher-level policy a skill policy, and the lower-level policy a (skill-conditioned) action policy.
- $z_s$ is called a "latent skill representation", while $z_T$ is called a "task embedding".  This (and other wording) makes two very similar concepts look different.  Call both of them "embeddings" to highlight the commonalities between skill-based and meta RL (and making their crucial differences stand out even better).
- The embeddings $z_s$ and $z_T$ are distinguished by their subscripts.  These subscripts are typeset like variables and are easily confused with states and targets, respectively, that use the same respective identifiers.  However, these subscripts are not variables but labels that distinguish the two different uses of $z$.  Typeset them as text (not math italic), make them superscripts to further distinguis them from variable subscripts ($z^{\textrm{s}}$, $z^{\textrm{T}}$) or even write them out: $z^{\textrm{skill}}$, $z^{\textrm{Task}}$.

  To further complicate things, in $z_T$ in Eqn. 2, $T$ is not a label but represents an actual training task (sampled from the distribution $p_T$ of training tasks, where $T$ again is a label, and characterized by its reward function $r_T$, where $T$ is this training task).

  One way to clarify this is the $z_{T, \textrm{target}}$ notation in Sec. 4.3.  Follow this or one of my suggestions throughout.

  This excessive overloading and abuse of notation makes this paper much harder to read than necessariy.
- In Eqns. 1, 2, 4 and 5 the argument of the summation includes the regularizer; a parenthesis around its two terms is missing.
- In Eqn. 2, the maximization should also run over the task encoder $q$ (and also over the critic, but the critic is anonymous here, so omitting it should not cause confusion).  Similarly in Eqn. 4(?).
- Below Eqn. 2, "skill" should read "task".
- In Eqn. 3, the plus should be a minus.
- The skill prior is conditioned on the starting state of a skill.  This is called $s_0$ in Fig. 2 and $s_1$ at the bottom of p. 4.

Some other issues that should be clarified:
- Do I understand correctly that in the Kitchen task, the only significance of a subtask is that the overall Task's reward function is structured to emit a reward after the completion of each subtask (in order)?
- In Fig. 4 and 6, SiMPL and PEARL first meta-train for 20 episodes and then train on the target task? Or what is the special significance of the 20th episode?

Another remark: It is bad news but unsurprising that SiMPL does not perform well if the training tasks are not representative of the target tasks. It would be highly interesting to try your method on Kitchen-like tasks where subtasks depend implicitly on other subtasks, without intermediate rewards.  In such hierarchical/combinatorial settings, there is no such immediate notion of "representative", and I can imagine that your method can (meta-)learn to combine skills to solve such hierarchical tasks.  Demonstrating this would be a big win.

**Summary Of The Paper:**

This paper combines skill-based with meta RL using training tasks.

The skill-based method (SPiRL, Pertsch et al. 2020) forms a skill embedding that represents $K$-step state-action trajectories.  An action policy provides an action distribution conditioned on the currently-active skill, and a (higher-level) skill policy is trained using RL with the skill embedding as its action space.  Simultaneously a skill prior conditioned on the starting state of a given skill is trained.  Once trained, the skill embedding, the skill-conditioned action policy, and the skill prior are held fixed.

The off-policy meta RL method (PEARL, Rakelly et al. 2019) forms a task embedding that represents small sets of skills active during a given task. (This is how I understand it; the paper does not explicitly state this. Rakelly et al. do not use skills; their task embedding represents small sets of state-action-state-reward sequences.)  A skill policy provides a skill distribution conditioned on the currently-active task.  Its training is regularized using the above skill prior. Once trained, the task embedding is held fixed.

Up to here, the system is trained in a task-agnostic manner.  To learn a target task, the system further trains the skill policy, conditioned on the target task, again regularized using the skill prior.

**Summary Of The Review:**

The paper proposes a nontrivial and effective way of combining skills extracted from offline training data with meta-learning from training tasks.  Following the details is hampered by notational ambiguities and inconsistencies.

Post-discussion edit: I upped my score to in response to the authors' improvements, although limited novelty remains a concern, as expressed by my fellow reviewers.

---

> ### Author Response · Authors · 2021-11-23
> **Response to Reviewer jZvC**
>
> We thank the reviewer for the thorough and constructive comments. Please find the response to your questions below.
>
> **Notation**: We thank the reviewer for the notational suggestions and corrections. We have updated our notations to make it clear.
>
> - $z_s$ and $z_T$ are replaced by $z$ and $e$, respectively. We believe it is more appropriate to use different identifiers for them.
> - We now use * superscript to denote “target”. Thus, $z_{T, target}$ is updated to $e^*$.
> - Following PEARL (Rakelly et al., 2019), we optimize the task encoder via the Q-function update during meta-training (see Appendix A.1.2 for the revised equation). Therefore, we do not include the task encoder in the policy update equation (4).
> - Multiple parentheses are missing. / Below Eqn. 2, "skill" should read "task". / In Eqn. 3, the plus should be a minus. / mismatch between s0 in Fig. 2 and s1 at the bottom of p. 4.: We thank the reviewer for pointing them out. We have fixed them all.
>
> **The significance of subtasks in the kitchen environment**: As mentioned by the reviewer, the agent receives a sparse reward signal upon completion of every subtask, but there is no further supervision provided to the agent, e.g. we do not train separate policies for each subtask, etc. In accordance with prior works (Gupta et al., 2019, Pertsch et al., 2020), we use the notion of “completed subtasks” to give a more intuitive estimate of the task progress in the quantitative evaluation, but as you suggested with your question there is no additional supervision provided to the agent through subtasks.
>
> **The initial 20 episodes for SiMPL and PEARL**: At the beginning of learning a target task, our proposed method and PEARL first collect 20 episodes of environment interactions, and then use them to condition the meta-trained policy before fine-tuning it further on the target task. Please note that these episodes are not used for further meta-training but only to condition the meta-test policy via the transition encoder q(e|c) (q(z_T | tau) in our initial submission ). This is the standard procedure used by PEARL for task adaptation during meta-test time and we apply the same approach in our method. Since these episodes count as online interactions with the target task environment, we include them in the quantitative evaluation of learning efficiency in Figures 4 and 6 for fair comparisons to other methods that do not use the initial 20 episodes for conditioning but directly learn from them.
>
> We hope that this addresses all the concerns raised in your review. Please kindly let us know if there are any further questions. We would be more than happy to address them. Thank you very much for all your detailed feedback and the time you put into helping us to improve our submission. If applicable, please consider raising your score accordingly.

---

### Official Review · Reviewer_AfGF · 2021-11-02

**Correctness:** 3
**Technical Novelty And Significance:** 2
**Empirical Novelty And Significance:** 2
**Recommendation:** 6
**Confidence:** 4

**Main Review:**

## Strengths

The paper is well written and the method is clearly explained. To my understanding it is a clean extension of SPiRL where the main point is to formulate the high level policy learning as a meta-RL problem.

* The meta-RL formulation allows to exploit available target task trajectories at test time efficiently. Indeed as shown in the experiments of Figure 4, when using 20 episodes to warm up the high-level policy, the initial success rate of SiMPL is ~2x higher than other methods after 20 rollouts on the target task environment.

* The meta-training task distribution analysis shown in Figure 6 is also an interesting experiment. Figure 6-(b,c) show the bias learned from the meta-trained dataset and the impact it has on the proposed method. This experiment is insightful and shows the advantage of the method if the train-test tasks are well aligned.

## Weaknesses
The method is close to SPiRL, e.g. it reuses the skill extraction from SPiRL and the main novelty comes from the skill-based meta-training part. It is unclear to me what is the real gain coming from the meta-RL method given the current experimental results which is either too restrictive or a bit unfair.

* The method seems to be tested in maze environments more restrictive than SPiRL, e.g. the train-test maze is the same, whereas SPiRL train on a various set of small mazes and test on larger mazes (c.f. Figure 3 left). Is the method trained on a fixed, pre-defined maze or does it work on a large set of mazes ? Could you evaluate the method on the same environments as SPiRL and compare to it ?

* The proposed method use extra annotations at test time which are target trajectories. These trajectories are not available to the strongest baseline SPiRL, so I wonder if the comparisons of Figure 4 are fair. A fairer experiment would be to also provide these extra annotations (target trajectories) to SPiRL. For example, you could add the available target trajectories to the replay buffer of SAC before running SPiRL. This way you could measure SPiRL vs SiMPL in a fair setup where both have access to the same amount of information.

* Overall the proposed method provides a good warm-start but eventually SiMPL and SPiRL methods seem to be converging to the same success rate after a limited number of steps. SPiRL seems already quite sample-efficient and the gains coming from SiMPL are a bit marginal after 200 episodes.

**Summary Of The Paper:**

The paper proposes a meta-RL method to learn to solve long-horizon tasks with sparse rewards efficiently. To do so it builds on previous works, mainly SPiRL, to learn a set of skills and skills prior from an offline dataset. Jointly with the skills, a high-level policy generating action in the learned latent space, is trained with meta-RL. The goal of this policy is to learn to compose learned skills to solve new tasks efficiently. Given a small set of target task trajectories, the policy can meta-learn from it and learn quickly to solve the target task. Experimentally the proposed method is shown to be more sample-efficient on two tasks, maze navigation and kitchen manipulation.

**Summary Of The Review:**

The paper is well-written and the method is clearly explained. The proposed method builds on SPiRL and proposed a meta-RL learning of the high-level policy. It shows reasonable improvements when the number of samples is limited (maze and kitchen tasks) and also provides insightful experiments (meta-training task distribution analysis).
However the performances are quikcly matched by concurrent baselines after not that many steps and the comparison to SPiRL seems unfair in the current setup. Mainly the set of mazes solved seem more restrictive than SPiRL, is it a restriction of the method? Also the extra annotations used by meta-RL during evaluation (target trajectories), is not made available to the strongest baseline the authors compare to, SPiRL. The experimental section could be strengthen by evaluating SiMPL against SPiRL where the extra-data provided to SiMPL is also provided to SPIRL, and by evaluating on a richer set of environments.

---

> ### Author Response · Authors · 2021-11-23
> **Response to Reviewer AfGF**
>
> We thank the reviewer for the thorough and constructive comments. Please find the response to your questions below.
>
> **Evaluation on SPiRL’s maze environment**: We thank the reviewer for this insightful suggestion. We have added an experiment on the original, image-based SPiRL maze environment where we use the original task-agnostic dataset published with the SPiRL paper, which was collected across a wide range of maze layouts, and then show efficient learning of new tasks in an unseen maze layout. We report results in Section C. We find that our method can perform well in this setting too and learns downstream tasks substantially faster than SPiRL.
>
> **Extra annotations at test time**: It seems that there was a misunderstanding about our method. We apologize for the confusion and will revise the paper to make it clearer. We would like to emphasize that our method does not use any extra annotations (e.g. target trajectories) during the target task learning phase (i.e. test time). Equivalently to PEARL, we use the meta-trained policy to collect a small set of “conditioning trajectories” but these trajectories have no further annotations, nor are they demonstrated trajectories for the target task. All other approaches (PEARL-ft, SAC, SPiRL, MTRL) have access to interactions with the target environment. We compare the total number of target task environment interactions in our quantitative comparisons (i.e. including the initial conditioning rollouts of our approach and PEARL). Thus, in summary, we believe the comparison is fair since our proposed methods, PEARL-ft, SAC, SPiRL, and MTRL have the same amount of information available on the target task. We will revise the paper to clarify this.
>
> **Efficiency gains**: The main benefit of the proposed approach and meta-learning methods is the fast learning of new tasks within a few episodes. Indeed, we find that early on in training our approach leads to substantially faster learning of the target task. To underline this advantage, we add evaluations where we condition our meta-trained policy on even fewer than 20 target task episodes (see Section B of the revised paper for the evaluations with 1, 2, 5, 10, ... episodes). We find that even with only 10 episodes of interaction with the target task in the maze environment, our method can solve 50% of the tasks. Similarly, in the kitchen environment, our method learns to solve two out of four subtasks using only 10 episodes, corresponding to only minutes of robot interaction time. Yet, the best-performing baseline needs an order of magnitude more experience to reach the same level of performance (see Figure 4). While the baselines can eventually attain comparable performance later in training, saving an order of magnitude on required interaction time is significant, especially if training is performed on real-robot systems where interactions with the environment are slow and costly.
>
> Please kindly let us know if there are any further concerns or missing experimental results that potentially prevent you from accepting this submission. We would be more than happy to address them. Thank you very much for all your detailed feedback and the time you put into helping us to improve our submission.

---

> > ### Comment · Reviewer_AfGF · 2021-11-24
> > **Thank you for the feedback**
> >
> > Evaluation on SPiRL's: Thanks a lot for adding a clear comparison to SPiRL. I have one question about Fig. 11 a) b) , all of the tasks presented are in a small subset of the maze. How come they are not spread across the maze ?
> >
> > Extra annotations: Ok I missed this while reading, thanks for making the meta-RL step clearer. I thus remove my comments about some experiments being potentially unfair, I agree there is no issue there and the proposed plots measures accurately the interactions with the environment of the different methods.
> >
> > Efficient gains: Agreed the gains are interesting, especially since there is no use of extra-annotations as I thought while writing the review.
> >
> > If you could please clarify the first point it would be great. Otherwise this is a strong rebuttal that should strengthen the paper and the proposed approach is interesting. I am inclined to update my grading.

---

> > > ### Author Response · Authors · 2021-11-24
> > > **Re: Thank you for the feedback**
> > >
> > > We thank the reviewer for acknowledging our rebuttal and additional experiments. Please find the response to your further questions below.
> > >
> > > **Comparison to SPiRL**:
> > > As suggested by the reviewer, to compare against SPiRL with the exact same setup used in [the SPiRL paper](https://arxiv.org/pdf/2010.11944.pdf). Specifically, we used the same initial position of the agent as well as sampled the tasks of comparable complexity to the ones used in the SPiRL paper for our evaluation (please see Figure 13 in the SPiRL paper for tasks used in their evaluation). Also please note that while the used test tasks do not fully cover the entire maze, they are already considerably long-horizon, requiring on average 710 steps until completion while only providing sparse goal-reaching rewards. If the reviewer feels it is needed, we can sample the tasks spreading across the entire maze and compare our proposed method and SPiRL, which we believe would yield consistent results.
> > >
> > > **Extra annotations and efficient gains**:
> > > We thank the reviewer for the clarification and for recognizing the performance gain of our proposed method.
> > >
> > > We hope that this addresses your concern. Please kindly let us know if there are any further questions or concerns. We would be more than happy to address them.

---

> > > > ### Comment · Reviewer_AfGF · 2021-11-27
> > > > **Re**
> > > >
> > > > I have updated my grade, thanks for the effort. The experimental section is in better shape now in my opinion.
> > > >
> > > > >  Also please note that while the used test tasks do not fully cover the entire maze, they are already considerably long-horizon, requiring on average 710 steps until completion while only providing sparse goal-reaching rewards. If the reviewer feels it is needed, we can sample the tasks spreading across the entire maze and compare our proposed method and SPiRL, which we believe would yield consistent results.
> > > >
> > > > As the goal of the paper is to solve long-horizon sparse-reward tasks I believe enlarging the set of target tasks distribution could be beneficial. If both SPIRL and your approach fail at some point it is also good to highlight it and exhibit their limitations. This would only strengthen the method analysis.

---

> > > > > ### Author Response · Authors · 2021-11-30
> > > > > **Re: Re**
> > > > >
> > > > > We thank the reviewer for the constructive feedback. We will compare our method and SPiRL by sampling the tasks spreading across the entire maze and include the result in the revised paper.

---

### Official Review · Reviewer_6qBE · 2021-11-02

**Correctness:** 4
**Technical Novelty And Significance:** 2
**Empirical Novelty And Significance:** 2
**Recommendation:** 6
**Confidence:** 4

**Main Review:**

The main strength of this work is its strong experimental results in the new setting it proposes.
- The proposed algorithm improves substantially over reasonably chosen baselines.
- Additionally, this work includes two relatively informative ablations which illustrate how the proposed approach performs as a function of number of meta-training tasks, and as a function of the alignment between the meta-training tasks and the meta-testing tasks.

Overall, there are no serious flaws with this work.

At the same time, the main contribution of this work is just a straightforward combination of two existing algorithms, SPiRL and PEARL. Considering and proposing the new setting of combining offline play data with the meta-RL setting is interesting, but feels a little incremental.

Additional minor comments:
- The notation in this work is difficult to read in places. For example, $z_T$ is used to denote the task embedding vector for PEARL, while $z_s$ is used to denote the skill variables. These are completely separate concepts, but the notation makes them seem related. Similarly, $\mathcal{T}$ is used to denote a set of tasks, which looks very similar to the trajectory $\tau$, even though they are completely different. I would recommend using more visually distinct notation.
- The PEARL-ft baseline seems somewhat strange to me. PEARL already prescribes the meta-test time behavior -- why should we change that to SAC instead? PEARL also seems to perform surprisingly poorly on its own, although it appears that this might just be due to the horizon length and reward sparsity. Approximately how many timesteps is each of these tasks?
- This work is missing citations to some of the foundational meta-RL works, such as [1, 2], as well as meta-RL works that do consider sparse reward tasks, such as [3, 4].

[1]: RL2: Fast Reinforcement Learning via Slow Reinforcement Learning. Duan et al., 2016.

[2]: Learning to reinforcement learn. Wang et al., 2016.

[3]: Decoupling Exploration and Exploitation for Meta-Reinforcement Learning without Sacrifices. Liu et al., 2021.

[4]: NoRML: No-Reward Meta Learning. Yang et al., 2020.

**Summary Of The Paper:**

This work considers a new setting where you can leverage both 1) offline "play" data that does not contain reward or task labels, and 2) meta-training tasks in order to quickly learn new tasks at meta-test time. To solve this setting, this work proposes to learn skills from 1) by using the SPiRL algorithm, and then learn a hierarchical policy on top of the learned skills over the meta-training tasks 2) using PEARL. Empirically, the combination of SPiRL and PEARL outperforms both SPiRL and PEARL on their own on a 2d maze navigation task and on a robot kitchen task.

**Summary Of The Review:**

Overall, I think this work is quite reasonable and does not have any serious issues. I find it only borderline on the accept side, due to its incremental nature.

---

> ### Author Response · Authors · 2021-11-23
> **Response to Reviewer 6qBE (2/2)**
>
> **Related work**: We thank the reviewer for suggesting the missing related works and have revised the paper to properly discuss them. The papers incorporated in the revised paper are as follows:
> - "RL2: Fast Reinforcement Learning via Slow Reinforcement Learning" in arXiv 2016 by Yan Duan et al.
> - "Learning to reinforcement learn" in CogSci 2017 by Jane X. Wang et al.
> - "Decoupling Exploration and Exploitation for Meta-Reinforcement Learning without Sacrifices" in ICML 2021 by Evan Zheran Liu et al.
> - "NoRML: No-Reward Meta Learning" in AAMAS 2019 by Yuxiang Yang, et al.
>
> We hope that this addresses all the concerns raised in your review. Please kindly let us know if there are any further questions. We would be more than happy to address them. Thank you very much for all your detailed feedback and the time you put into helping us to improve our submission. If applicable, please consider raising your score accordingly.

---

> ### Author Response · Authors · 2021-11-23
> **Response to Reviewer 6qBE (1/2)**
>
> We thank the reviewer for the thorough and constructive comments. Please find the response to your questions below.
>
> **Novelty**: The main contribution of our work is the usage of task-agnostic experience to enable meta-learning on sparse, long-horizon tasks. We would like to emphasize that while our method builds on prior work in skill-based RL and meta-learning, our proposed method involves several novel techniques that are essential for enabling stable meta-learning with learned skills:
>
> - **Residual policy learning (see Section D.1.3)**: we parameterize our policy to predict the residual, i.e. the difference, with respect to the pre-trained skill prior instead of directly outputting the skill distribution. The residual parametrization stabilizes learning by ensuring that the initial skill distribution is close to the skill prior (note that we cannot simply initialize the policy to the skill prior, as done in Pertsch et al., 2020, since we additionally need to condition the policy on the context vector c).
> - **Adaptive policy regularization (see Section 4.2)**: we train our approach with variable sizes of conditioning transition sets. Since the amount of task information that is contained in this set can vary widely depending on the set’s size, particularly in sparse reward tasks, our method uses varying target KL divergence values for regularizing the policy towards the prior (stronger regularization for smaller transition sets), while prior works apply regularization with constant weight (Rakelly et al., 2019).
> - **Transfer of meta-trained Q-function (see Section D.2.2)**: together with the meta-trained policy, the Q-function and KL regularization coefficient are also transferred to the target task learning phase. This warm start of the Q-function prevents the policy from instantly falling back to the skill prior, which happens when using a randomly initialized Q-function and $alpha_0$ = 1 as in prior work (Pertsch et al., 2020).
>
> Without these techniques, we were unable to reliably meta-train skill-based policies, especially on the harder kitchen environment tasks. We believe that these techniques have not been employed in the aforementioned prior works and thus, together with the problem setting, constitute the novelty and technical contributions of our work. While the original submission already detailed these techniques, we have now revised the paper to underline their importance for stable meta-training in Section 4 and detail them in Section D.
>
> **Notation**: We thank the reviewer for pointing out the confusion. We have updated the notation in the revised paper to make it clear.
> - $z_s$ and $z_T$ are replaced by $z$ and $e$, respectively.
> - $\tau$ is replaced by $c$.
>
> **PEARL-ft**: As mentioned by the reviewer, the original PEARL paper does not fine-tune at test time. In our experiments, we follow the same test-time procedure proposed in the original PEARL paper to obtain the performance of PEARL. Additionally, we further fine-tune the PEARL model, dubbed PEARL-ft, using online environment interactions to allow for a fair comparison with other approaches (ours, SPiRL, SAC, MTRL), which also have access to further interactions for fine-tuning. We empirically found that fine-tuning PEARL indeed improves its target task performance, but it is still significantly less sample efficient compared to methods that leverage offline datasets. In Figure 4, the performance of the original PEARL is visualized with dotted purple curves and the performance of the fine-tuned PEARL is visualized with purple curves.
>
> **PEARL’s performance**: We empirically find that PEARL (with and without fine-tuning) struggles on sparse, long-horizon tasks since the exploration during meta-training struggles to obtain any rewards. To closely investigate this, we include new experiments with varying task horizons in the maze navigation domain (see Section A). The experimental results suggest that PEARL can solve short-horizon maze tasks (see Figure 8a) but struggles at solving more challenging, long-horizon maze tasks (Figure 8(b) and Figure 8(c)) due to insufficient exploration (see Figure 9 for the qualitative results of PEARL rollouts). Similarly, in the kitchen environment, a successful task execution requires approximately 200 steps, which is also substantially longer than the tasks tested in the original PEARL paper. We believe the long task horizon and the sparse-reward feedback constitute major challenges to prior meta-RL algorithms like PEARL, yielding the poor performance.

---

> > ### Comment · Reviewer_6qBE · 2021-11-25
> > **Reply**
> >
> > Thank you for the clarifications and the updates to the paper! I no longer have any concerns about the technical, related work, and clarity aspects of the paper.
> >
> > Regarding the novelty aspects, I appreciate that the authors have highlighted technical novelties in the rebuttal. However, the significance of these technical novelties is somewhat unclear to me (though I have understood that the proposed approach does not work without them!):
> > - **Residual policy learning**: I don't quite understand why the residual parametrization helps keep the output distribution close to the skill prior. If the policy outputs a large residual, then it can still get far away from the prior? If the goal is to stay close to the prior, why not regularize with something more principled, e.g., a KL term (which is something this work is already doing?)?
> > - **Adaptive policy regularization**: This seems fair, but adapting hyperparameters via dual gradient descent (as described in Section 4.2) is a fairly standard technique, e.g., also seen in "Variational Discriminator Bottleneck: Improving Imitation Learning, Inverse RL, and GANs by Constraining Information Flow." Separately, Section 4.2 appears to be somewhat contradictory in how $\alpha$ is tuned. Initially, it states that it's tuned via dual gradient descent (here, it may be useful to clarify exactly what constrained optimization problem dual gradient descent is performed on. I understand that it's defined in the Pertsch paper, but it's not possible to understand this without reading that paper). Later on, the bottom of Section 4.2 says that $\alpha$ is just chosen between two values $\alpha_1$ and $\alpha_2$. Can you clarify this?
> > - **Transfer of the meta-trained Q function**: Sorry, I'm slightly confused here. Isn't the default thing to do using the meta-trained Q-function instead of randomly initializing a new one?
> >
> > Overall, these appear to be (key) implementation details rather than critical parts of the high-level approach, which is reflected in the fact that 2 of the 3 of them appear in the Appendix, rather than in the main body. I don't think there's anything wrong with that, but I'm not sure that it improves the technical novelty of the paper either.
> >
> > Altogether, I still believe the novelty in this work is limited, which reviewer 1ZNY agrees on. However, despite all of the above discussions about novelty, I don't necessarily think that novelty is so important, especially given its subjective nature. I do think the empirical contributions of the work put it over the bar for acceptance. Hence, I'm overall inclined to keep my score at a 6.

---

> > > ### Author Response · Authors · 2021-11-27
> > > **Re: Reply**
> > >
> > > We thank the reviewer for reading our rebuttal. Please find the response to your further questions below.
> > >
> > > - **Residual policy learning**: As mentioned by the reviewer, optimizing the KL term encourages the output distribution to stay close to the prior distribution. Additionally, with the residual parameterization, it is easier for the policy to optimize the KL term since predicting standard normal distribution N(0, I) would yield the prior distribution. In other words, the residual parameterization is an inductive bias that we employ to make optimizing the KL term easier.
> > > - **Adaptive policy regularization**: We apologize for the confusion. We adaptively tune the KL regularization coefficient $\alpha$ to regularize the policy depending on the size of the encoding set. Specifically, we use two $\alpha$ values, and corresponding $\delta$ values, for both the two domains -- for the maze navigation domain, we set $\delta=0.1$ for the batch that is conditioned on size 4 transitions and $\delta=0.4$ for batch conditioned on size 8192 transitions and for the kitchen manipulation domain, we set $\delta=0.4$ for bath conditioned with a size 1024 transitions while $\alpha=0.3$ for batch conditioned on size 2 transitions (see Appendix D.2.1).
> > > - **Dual gradient descent optimization objective**: We thank the reviewer for the suggestion. We will revise the paper to explicitly state what constrained optimization problem the dual gradient descent is performed on.
> > > - **Transfer of the meta-trained Q function**: We agree with the reviewer that this part can be intuitive in meta-RL. Yet, it is not a common practice in skill-based RL, which is why we emphasize it. Based on the reviewer's suggestion, we will revise the paper to tone down this point. We will introduce the meta-trained Q function transfer as a key procedure of our approach and discuss relevant work rather than describe it as a contribution.
> > >
> > > We hope that this addresses your questions. Please kindly let us know if there are any further questions or concerns. We would be more than happy to address them.

---

> > > > ### Comment · Reviewer_6qBE · 2021-11-30
> > > > **Reply**
> > > >
> > > > Great, thanks for the clarification. Overall, I still find the empirical contribution to be compelling, even if the novelty is limited, and still recommend acceptance with a score of 6.

---

### Author Response · Authors · 2021-11-23
**Author Response Summary**

We would like to sincerely thank all the reviewers for their thorough and constructive comments. We have revised our paper to incorporate them. The major changes are summarized as follows.

**Additional experiments**: We have conducted the following experiments to better answer the questions asked by the reviewers. The results and the corresponding discussions have been included in the revised paper (Section A, Section B, and Section C).
- **Section A - Meta-RL Method Ablation (BC+Meta-RL methods)**: this experiment evaluates meta-learning with a behavioral cloning (BC) policy learned from the offline dataset on three task distributions with increasing difficulty. This experiment also investigates the performance of our method and the baseline with varying horizons. The result highlights the importance of learning temporally extended skills from the offline datasets. (Reviewer 1NZY)
- **Section B - Learning Efficiency on Target Tasks with Few Episodes of Experience**: this experiment examines the data efficiency of the compared methods on the target tasks, specifically when provided with only a few (<20) episodes of online interaction with an unseen target task. The result highlights the sample efficiency of the proposed method. (Reviewer AfGF)
- **Section C - Investigating Offline Data vs Target Domain Shift**: this experiment investigates the performance of the proposed method in the original maze navigation environment of Pertsch et al., 2020 (SPiRL) which features image-based observations and substantial domain shift between offline data and the target domain. The results show that our approach is robust to such domain shifts and still improves sample efficiency substantially. (Reviewer AfGF, Reviewer 1NZY)

**Novelty**: The main contribution of our work is the usage of task-agnostic experience to enable meta-learning on sparse, long-horizon tasks. We would like to emphasize that while our method builds on prior work in skill-based RL and meta-learning, our proposed method involves several novel techniques that are essential for enabling stable meta-learning with learned skills. We believe that these techniques have not been employed in the aforementioned prior works and thus, together with the problem setting, constitute the novelty of our work. We have revised the paper to explicitly mention the importance of these techniques in Section 4 and point to their detailed discussion in Section D. (Reviewer 6qBE, Reviewer 1NZY)

**Related work**: We thank the reviewers for pointing out relevant works that are missing and we have revised the related work section to include the following references and properly discuss them:

- "RL2: Fast Reinforcement Learning via Slow Reinforcement Learning" in arXiv 2016 by Yan Duan et al. (Reviewer 6qBE)
- "Learning to reinforcement learn" in CogSci 2017 by Jane X. Wang et al. (Reviewer 6qBE)
- "Decoupling Exploration and Exploitation for Meta-Reinforcement Learning without Sacrifices" in ICML 2021 by Evan Zheran Liu et al. (Reviewer 6qBE)
- "NoRML: No-Reward Meta Learning" in AAMAS 2019 by Yuxiang Yang, et al. (Reviewer 6qBE)
- "Fast Context Adaptation via Meta-Learning" in ICML 2019 by Luisa M Zintgraf et al. (Reviewer 1NZY)

**Details**: We have revised the main paper and Appendix to clearly organize the information related to the topics listed below.
- The description of the PEARL-ft baseline (Reviewer 6qBE)
- The significance of subtasks in the kitchen environment (Reviewer jZvC)
- The initial 20 episodes for SiMPL and PEARL (Reviewer jZvC)

**Notations**: We have updated the notation in the revised paper following the reviewers’ suggestions. We appreciate the reviewers’ feedback and corrections. (Reviewer 6qBE, Reviewer jZvC)

---

### Decision · Program_Chairs · 2022-01-20

**Decision:**

Accept (Poster)

**Comment:**

The main contribution of this paper lies in the novel setting that is being considered: offline data without rewards is combined with meta-training tasks to quickly adapt to new long-horizon tasks at meta-test time. Within this setting, it is shown that the combination of SPiRL and PEARL outperforms the individual algorithms. The technical contribution is limited, as now new methods are introduced. Nevertheless, the setting considered is interesting and the empirical evaluation is solid. For these reasons, I recommend acceptance.